# Super-resolution imaging of native fluorescent photoreceptors in chytrid fungal eyes

Wayne Busse[1], Enrico Klotzsch [ID] [2,3,4], Yousef Yari Kamrani [ID] [1], Natalie Wordtmann [ID] [1], Simon Kelterborn [ID] [5], Peter Hegemann [ID] [1] & Matthias Broser [ID] [1✉]

## Abstract

**Photoorientation in motile fungal zoospores is mediated by rhodopsin guanylyl cyclases (RGCs). In certain chytrids, these photoreceptors form heterodimers consisting of a visible-light-absorbing RGC paired with neorhodopsin (NeoR), a rhodopsin distinguished by its unique spectral properties: far-red absorption and high fluorescence. Leveraging the native fluorescence of NeoR, we detected RGCs in living zoospores of the fungus *Rhizoclosmatium globosum*. The reversible photoswitching of bistable NeoR enabled super-resolution microscopy, facilitating single-molecule detection and quantification of NeoR proteins within individual zoospores. This approach also revealed the precise localization of RGCs within the rumposome, a chytrid-specific organelle hypothesized to mediate photoreception. Fluorescence tracking across different stages of the chytrid life cycle and the analysis of transcriptomic data confirmed that RGCs are predominantly present during the zoospore stage. Functional assays of recombinantly expressed RGC heterodimers with modified substrate specificity revealed that only one of the two pseudo-symmetric nucleotide-binding sites is catalytically active. Strikingly, disrupting nucleotide binding in the non-catalytic site enhanced light-triggered cyclase activity by up to ninefold, indicating an allosteric regulatory mechanism in heterodimeric RGCs.**

**Keywords** Super Resolution Microscopy; Fungal Zoospores; Guanylyl Cyclase; Light Perception; Microbial Rhodopsin
**Subject Categories** Microbiology, Virology & Host Pathogen Interaction; Signal Transduction

## Introduction

Phototaxis in eukaryotic microorganisms relies on a finely tuned interplay between light perception, signal transduction, and flagellar movement within a single cell. A well-studied example is the pronounced phototaxis in the unicellular green alga *Chlamydomonas reinhardtii*, mediated by light-activated cation and proton channels named channelrhodopsins (Nagel et al, 2002; Nagel et al, 2003). These retinylidene photoreceptors are localized within a pigmented eyespot that partially overlaps with the plasma membrane (Krimer et al, 2023). In *Euglena gracilis*, phototaxis depends on heterodimeric photoactivated cyclases containing flavin chromophores, which work in conjunction with cAMP-sensitive channels at the flagellar base (Iseki et al, 2002; Häder et al, 2017). In both organisms, these photoreceptors are embedded within directional optical machinery that facilitates light detection and orientation.

In 2014, Avelar and colleagues characterized the first rhodopsin guanylyl cyclase (RGC) in an eyespot-like organelle in zoospores from the aquatic fungus *Blastocladiella emersonii* (*Be*RGC), a member of the phylum Blastocladiomycota (Avelar et al, 2014). RGCs are fusion proteins composed of a microbial rhodopsin and a C-terminally coupled class III cyclase domain (Mukherjee et al, 2019). These photoreceptors function as homodimers. Green-light absorption induces isomerization of the all-*trans* retinal chromophore, triggering conformational changes that reorient the cyclase domains to enable the conversion of GTP to the second messenger cGMP. While it was shown before that fungal phototaxis relies on rhodopsin photoreceptors, the identification of *Be*RGC, along with its activation of a downstream cGMP-gated potassium channel, provided first insights into the underlying signaling process (Saranak and Foster, 1997; Avelar et al, 2014; Avelar et al, 2015). Furthermore, the discovery of RGCs paved the way for their use as optogenetic tools to manipulate cGMP pathways through light, although many aspects of the molecular mechanism of light-induced cyclase activation remain unclear (Gao et al, 2015; Scheib et al, 2015; Scheib et al, 2018).

Facilitated by the increase of available genomic data, a second type of RGC that functions as heterodimer was identified in fungi belonging to the early diverged phylum Chytridiomycota (Broser et al, 2023). Heterodimeric RGCs, first characterized from *Rhizoclosmatium globosum*, are composed of a conventional visual-light-absorbing rhodopsin responsible for activation of the cyclase that pairs with an unusual far-red-absorbing rhodopsin, termed neorhodopsin (NeoR), with peak absorption close to

[1]Institute of Biology, Experimental Biophysics, Humboldt-Universität zu Berlin, Invalidenstrasse 42, 10115 Berlin, Germany. [2]Institute of Biology, Mechanobiology, Humboldt-Universität zu Berlin, Invalidenstrasse 42, 10115 Berlin, Germany. [3]Berlin Institute of Health (BIH) at Charité – Universitätsmedizin Berlin, BIH Center for Regenerative Therapies (BCRT), Augustenburger Platz 1, 13353 Berlin, Germany. [4]Experimental and Clinical Research Center, Max Delbrueck Center for Molecular Medicine and Charité Universitätsmedizin, Augustenburger Platz 1, 13353 Berlin, Germany. [5]Berliner Hochschule für Technik (BHT), Luxemburger Straße 10, 13353 Berlin, Germany. ✉E-mail: matthias.broser@hu-berlin.de

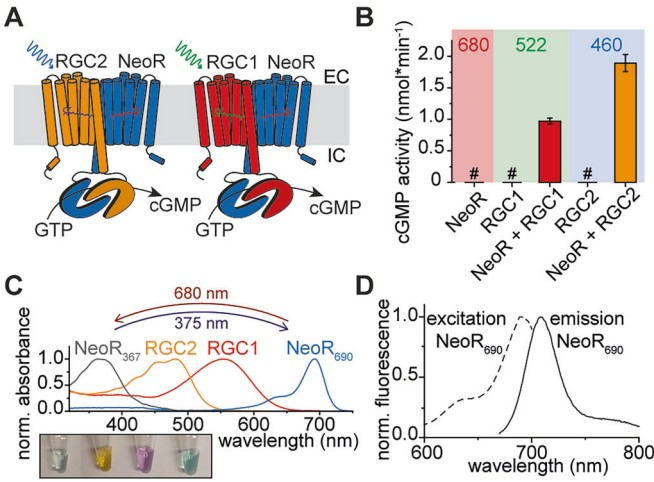

**Figure 1. Overview of rhodopsin guanylyl cyclases found in *R. globosum*.**

(A) Illustration of both heterodimeric rhodopsin cyclases found in *R. globosum*. Light excitation of RGC1 or RGC2 activates the enzyme resulting in cGMP-synthesis. (B) Light-induced cyclase activity tests of RGCs (at 680 nm, 522 nm or 460 nm) (co-)expressed in HEK293T cells, quantified via reverse-phase HPLC. $n = 3$ (technical replicates), mean ± S.E., # = not detectable. (C) Absorption spectra of recombinantly expressed and detergent-purified *R. globosum* RGCs (rhodopsin domains). $NeoR_{690}$ was photoconverted to $NeoR_{367}$ with 680 nm illumination and recovered using 375 nm light. Inset shows purified protein samples arranged in the same order as the spectra shown above. (D) Excitation (dashed) and emission (solid) spectra of $NeoR_{690}$ (black). Source data are available online for this figure.

700 nm (Fig. 1). NeoR is a bistable photochromic photoreceptor that can be switched between a far-red-absorbing ($NeoR_{690}$) and a UV-state ($NeoR_{367}$) (Fig. 1C), thereby modulating the light-response of the heterodimeric complex. However, the underlying functional mechanism is unknown (Broser et al, 2020).

The NeoR far-red absorbing state is highly fluorescent (Fig. 1D) with a fluorescence quantum yield (QF) of 20% observed for $RgNeoR_{690}$ from *R. globosum*. Together with their high extinction coefficient, neorhodopsins are currently the brightest fluorescent proteins in the far-red spectral region, where biological tissue is most transparent, surpassing even engineered fluorescent proteins for deep tissue imaging (Appendix Table S1). Both the far-red absorption and the high fluorescence are unique among retinal chromophores, and even fluorescence-optimized rhodopsins intended to serve as voltage sensors barely achieve a QF beyond 1% (McIsaac et al, 2014). Nevertheless, the intrinsic fluorescence of NeoR has not yet been exploited. The ability to switch NeoR between a far-red fluorescent state and a non-fluorescent and UV-absorbing state renders it a candidate for single-molecule localization microscopy (SMLM) (Klotzsch et al, 2015; Lelek et al, 2021).

So far, heterodimeric rhodopsins have been limited to RGCs. In *R. globosum*, NeoR alternatively forms functional heterodimers with either RGC1 ($\lambda_{max} = 550$ nm) or RGC2 ($\lambda_{max} = 480$ nm) (Broser et al, 2020) (Fig. 1). In contrast to rhodopsins, heterodimerization is well known for class III cyclases, especially those of higher eukaryotes. In these cyclases, the domains form two pseudo-symmetric active sites at the dimerization interface with often distinct functional roles (Steegborn, 2014). For example, in mammalian membrane-bound adenylyl cyclase (tmAC), a pseudo-heterodimer encoded within one polypeptide, the degenerate

non-catalytic site strongly enhances the enzyme activity of the remaining activate site after binding the diterpene forskolin. However, for heterodimeric RGCs the catalytic turnover and allosteric control are unknown.

In this work, we pursued the characterization of heterodimeric RGCs in their original host *Rhizoclosmatium globosum*. We confirmed the transcription of all RGC-encoding genes and exploited NeoR's intrinsic fluorescence to showcase single-molecule localization microscopy (SMLM) and to localize the photoreceptor in living zoospores. Additionally, we investigated the mechanism of heterodimeric RGC activation, thereby clarifying the involvement of the two putative substrate-binding sites in catalysis.

# Results

## RGC-encoding genes are individually transcribed in *R. globosum*

All three RGC-encoding genes of *R. globosum* (*neor*, *rgc1*, and *rgc2*) are clustered in the genome with *rgc1* and *rgc2* sequentially arranged as tandem, while *neor* is coded by the complementary strand (Fig. 2A) (Broser, 2022). Such an arrangement is common for heterodimeric RGC genes found in various chytrids and may allow a concerted bidirectional transcription from a single transcription start region (Xu et al, 2009; Broser et al, 2023). To assess RGC transcription in *R. globosum*, we harvested zoospores, extracted the mRNA and constructed a complementary DNA (cDNA) library by reverse-transcription PCR (RT-PCR). Since chytrid zoospores are transcriptionally inactive, the detected RNA likely originates from maternally inherited and inactive mRNA-loaded ribosomes (Medina and Buchler, 2020). Using specific primers (Appendix Table S2) that result in amplification of ~150–400 bp segments, we were able to confirm transcription of all three RGC-encoding genes (Fig. 2B). To verify the correct DNA sequence, we amplified and sequenced longer fragments of the N-terminal gene regions (Fig. 2; Primer: 1 + 4; 5 + 8; 9 + 12). Since *rgc1* and *rgc2* are arranged sequentially in the genome, we further tested for the presence of polycistronic mRNA of the two genes, a feature typically associated with prokaryotic transcription. The lack of amplification products suggests that each gene is transcribed individually.

## Expression of the fluorescent photoreceptor NeoR in *R. globosum*

Next, we employed fluorescence spectroscopy to detect intrinsic NeoR fluorescence in living zoospores. Zoospores were concentrated, and excitation and emission spectra were measured (Fig. 2C). Both the emission spectrum peaking at 708 nm and the excitation spectrum ($\lambda_{max} = 689$ nm) with a shoulder around 640 nm closely matched that of recombinant expressed NeoR in detergent. Long-term far-red illumination (60 min) with 680 nm significantly decreased the fluorescence. However, more than 80% of the initial fluorescence was recovered upon UVA light application (375 nm), confirming the bistable nature of NeoR. As no exogenous retinal was supplied during cultivation, this further demonstrated that the fungus synthesizes its own chromophore, which is consistent with previous reports identifying essential genes for retinal biosynthesis

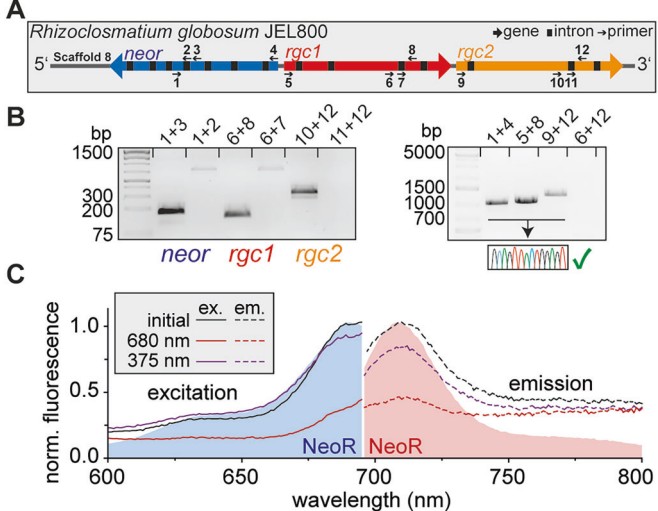

**Figure 2.  RT-PCR of RGC-encoding genes and in vivo fluorescence spectroscopy.**

**(A)** Genomic arrangement of all RGC-encoding genes in *R. globosum* JEL800 with primer binding sites used for reverse-transcription PCR. **(B)** Amplification products on cDNA synthesized from extracted zoospore RNA with primer used as shown in (A) (Appendix Table S2). **(C)** In vivo excitation and emission spectra of concentrated *R. globosum* zoospores ($OD_{600} = 2$), overlaid with spectra of detergent purified NeoR rhodopsin domain. Spores were photo-bleached with 680 nm illumination for 60 min, followed by recovery with 375 nm illumination for 1 min. Source data are available online for this figure.

(Galindo et al, 2022). The spectral properties of NeoR indicate the presence of all-*trans* retinal as the native chromophore, excluding other retinal variants such as 3,4-dehydro-retinal (occurring in some aquatic vertebrates), which would shift the absorption to ~760 nm (Broser et al, 2020).

## NeoR is localized in the fungal eye

To examine individual zoospores, we used a spinning disc confocal microscope. Cells were embedded in low-melting agarose to diminish motility. The spore cell body is ~4 µm in diameter with a single ~20 µm flagellum attached (Fig. 3A). A roughly 1 µm lipid-rich globule is the most clearly distinguishable organelle. Previous transmission electron micrographs of sliced zoospores revealed that the lipid globule is associated with an electron-dense microbody and a tubular structure, termed rumposome, localized in the space between the globule and the plasma membrane (Barr and Hartmann, 1976; Laundon et al, 2022). This subcellular structure, first described as microbody-lipid globule complex (MLC; Powell 1976), has been discussed as light perception machinery in fungal zoospores for over five decades (Chambers et al, 1967; Kazama, 1972; Barr and Hartmann, 1976).

Indeed, we observed NeoR fluorescence as a crescent-shaped subcellular structure between the lipid globule and the cell membrane (Fig. 3A, 640 nm). Staining the globule with NileRed (Fig. 3A, 560 nm), a lipophilic fluorescent probe, confirmed that NeoR is adjacent to the lipid droplet, indicating its localization in a structure within the posterior region of the spore and oriented towards the flagellum. Applying 640 nm laser light enables the

photoconversion of NeoR in individual spores ($n = 4$), which led to a decrease in fluorescence of more than 85% (Fig. 3B). Subsequent illumination of the same zoospore with a 405 nm pulse recovered >80% of the initial fluorescence, confirming the bistability of NeoR in its native organelle in accordance with our fluorescence spectroscopy data.

The reversible switching between a far-red fluorescent state and a UVA-absorbing non-fluorescent state enabled the application of Single-molecule localization microscopy (SMLM) to quantify protein levels and enhance imaging resolution (Fig. 3C, I, II). Using a custom-built super-resolution microscope, individual photons were detected post-bleaching by counting blinking events, which resulted from the low-dose UV-induced reconversion of NeoR fluorophores (Fig. EV1, see Methods for details) (Klotzsch et al, 2015). These photon bursts from individual fluorophores were subsequently analyzed to estimate the total number of NeoR proteins, yielding an average of $3417 \pm 1543$ per zoospore ($n = 23$) (Fig. 3C, III) (Ries, 2020). We further observed a periodicity of fluorescence intensity along the rumposome and identified a repetitive distance of ca. 30 nm based on spatial autocorrelation analysis (Fig. EV1). This finding aligns with prior transmission electron microscopy images, which revealed that the rumposome is composed of numerous hexagonal arranged tubes with a center-to-center distance of approximately 39 nm, that form a honeycomb-like structure (Kazama, 1972). To our knowledge, this is the first naturally occurring photosystem that can be measured in vivo, with fluorescence microscopy providing structural detail well below conventional resolution limits.

## Heterodimeric RGC numbers peak in the zoospore stage

Saprophytic chytrids exhibit a dimorphic life cycle, alternating between a motile zoospore and a sessile evolving form. Upon settling, the zoospore differentiates into a germling, which then develops into an immature thallus by forming a rhizoid system and an apophysis. The final stage of asexual reproduction is a sporangium that produces numerous new zoospores (Berger et al, 2005). We tracked the NeoR fluorescence across the four previously defined stages of the chytrid life cycle: zoospore, germling, immature thallus and sporangium.

In addition to the zoospore, we detected NeoR fluorescence during the germling stage and found multiple fluorescent spots in individual sporangia, corresponding to developing zoospores (Fig. 3E). However, no fluorescence was observed in the immature thallus, which is consistent with prior reports describing the degradation of the eyespot-like organelle at this stage (Laundon et al, 2022).

Laundon and colleagues have constructed an RNA library from synchronized culture samples of *R. globosum*, capturing zoospores, germling, immature thallus, and a mixed culture of unsynchronized cells (Laundon et al, 2022). We reanalyzed their dataset to assess RGC expression (Appendix Table S3). Normalized RGC transcript levels as shown in Fig. 3D peaked in the zoospore stage and were also prominent in the mixed culture, but they were lowest in the immature thallus, aligning closely with our fluorescence micro-scopy observations. Notably, in zoospores, the *neor* mRNA amount was at the highest level among RGC genes, followed by *rgc1* and *rgc2* transcripts with the latter showing a substantially lower level. The strongly reduced quantity of RGC transcripts in the germling

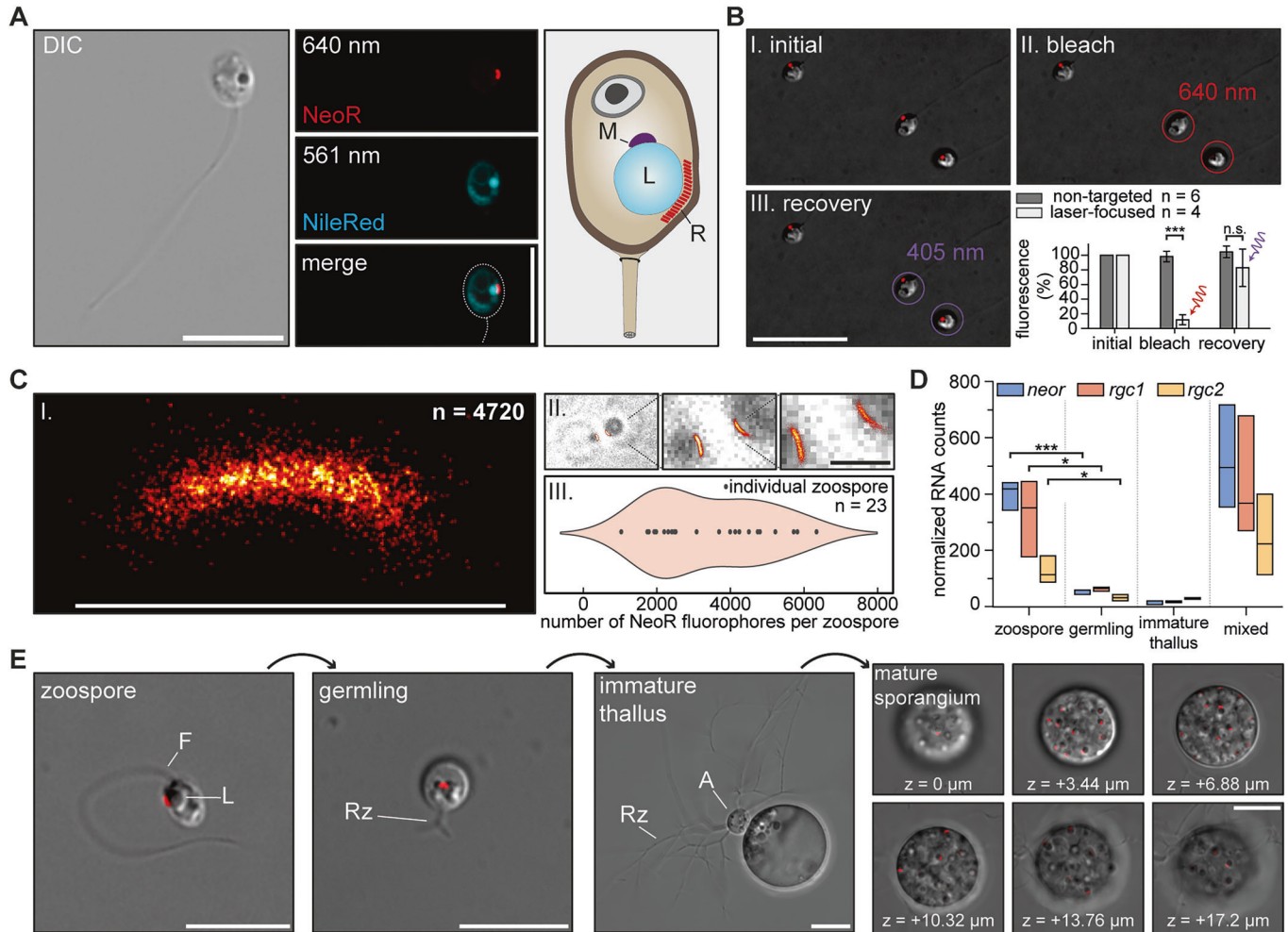

**Figure 3. Fluorescence microscopy of living *R. globosum* zoospores.**

(A) Differential interference contrast (DIC) and fluorescence micrographs of *R. globosum* zoospores. Freshly harvested zoospores were embedded in 0.25% low-melt agarose and lipids were stained with NileRed. The illustration of the microbody-lipid globule complex (MLC) was adapted from prior studies (Laundon et al, 2022). (B) Representative bleaching experiment on zoospores using a focused laser excitation. Total experiment with $n = 6$ untreated and $n = 4$ treated cells (biological replicates) are summarized in the bar chart. mean ± S.D., unpaired t-test left-to-right: ***$p = 4 \times 10^{-6}$, $p = 0.24$ (n.s. = not significant). (C) Single-molecule localization microscopy of zoospores. (I) The representative high-resolved structure represents the detection of $n = 4720$ NeoR molecules in a single zoospore. (II) Zoom-in of two zoospores with single NeoR molecule resolved fluorescence. (III) Violin plot displaying the distribution of fluorophore counts across 23 zoospores. The observed variation likely arises from differences in zoospore maturation under slightly varying microenvironments. (D) Box plot of normalized RNA-sequencing counts for individual RGC genes, based on transcriptome analysis of data previously published (Laundon et al, 2022). RNA was extracted from each stage of the *R. globosum* life cycle. Box plot displays the minimum, median (center line), and maximum values. $n = 3$ (biological replicates), unpaired t-test left-to-right: ***$p = 3 \times 10^{-4}$, *$p = 0.03$, *$p = 0.03$. (E) Life cycle of *R. globosum*, shown with merged DIC and 640 nm excitation micrographs. For sporangium, representative z-Stack images in 3.44 µm intervals are shown. For (A), (B), (E): scale bar = 10 µm; (C): scale bar = 1 µm. Source data are available online for this figure.

stage, despite detectable NeoR fluorescence, suggests that NeoR is not actively transcribed in the germlings and the detected NeoR protein is a leftover from the zoospore stage.

We also explored the transcription of two putative cGMP-gated ion channels (Appendix Fig. S1) that were hypothesized to mediate electric signaling in response to RGC light activation. Given the lack of a cell wall in zoospores (Medina and Buchler, 2020), we attempted to measure photocurrents by using suction pipettes as previously done for *C. reinhardtii* (Baidukova et al, 2022). However, the small size and rigid structure of the zoospores prevented the formation of an adequate electric seal between cell and pipette, hindering the study of the proposed pathway.

## Structural basis and enzymatic properties of the NeoR/RGC1 heterodimer

We used AlphaFold3 to predict the structural arrangement of the NeoR/RGC1 heterodimer. The received model revealed the typical architecture of RGCs, comprising of a rhodopsin (Rh) domain coupled to a catalytic guanylyl cyclase (GC) through a coiled-coil (CC) linker (Fig. 4A). The GC domain adopts the characteristic fold of type III cyclase with an antiparallel arrangement of the monomers that form two symmetric (in homodimers) or pseudo-symmetric (in heterodimers) nucleotide-binding sites at the dimerization interface (Steegborn 2014; Scheib et al, 2018; Butryn

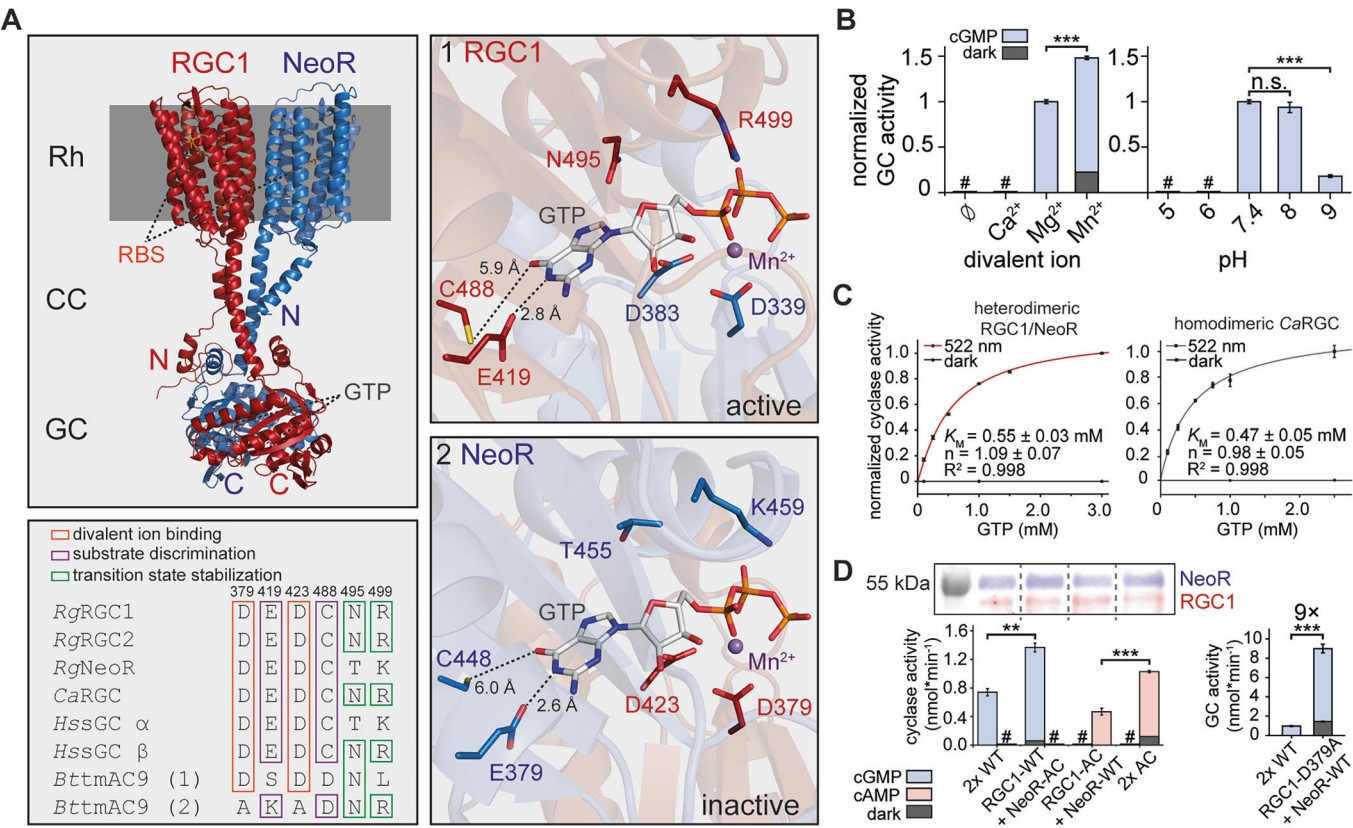

**Figure 4. Functional enzyme assays with full-length heterodimeric RGCs.**

(A) Overall arrangement of heterodimeric NeoR (blue)/RGC1 (red), as predicted by AlphaFold3 (Abramson et al, 2024). GTP-bound catalytic centers: (1 RGC1) with base-discriminating residues (C488/E419) provided by the RGC1 subunit; (2 NeoR) with C448/E379 from the NeoR subunit. Note that the metal binding aspartates are provided by the opposite protomer. RBS retinal binding site, GTP Guanosine-5'-triphosphate. Sequence alignment of crucial amino acids in the catalytic center of various class III cyclases (Rg - Rhizoclosmatium globosum; Ca - Catenaria anguillae; Hs - Homo sapiens; Bt - Bos taurus) with RgRGC1 numbering. (B) Enzymatic activity screen of RGC1/NeoR heterodimer in crude membrane HEK293T fractions with different divalent ions (5 mM) or pH and 2 mM GTP. Cyclase activity was measured under light activation (522 nm) or in darkness. $n = 3$ (technical replicates), mean ± S.E., # = not detectable, unpaired t-test left-to-right: ***$p = 7 \times 10^{-7}$, ***$p = 9 \times 10^{-7}$, $p = 0.3$ (n.s. = not significant). (C) Light-triggered enzyme activity (Hill fit) of NeoR/RGC1 and homodimeric CaRGC determined in crude membrane fractions with 10 mM MgCl₂. $n = 3$ (technical replicates), mean ± S.E. (D) Immunoblot of wild-type RGC1/NeoR and adenylyl-cyclase (AC) variants expressed in HEK293T cells. Cyclase activity of each AC-variant and the RGC1-D379A mutant. Heterodimeric constructs were normalized using intrinsic NeoR-fluorescence (see Methods for details). $n = 3$ (technical replicates), mean ± S.E., # = not detectable, unpaired t-test left-to-right: **$p = 1 \times 10^{-3}$, ***$p = 4 \times 10^{-4}$, ***$p = 5 \times 10^{-5}$. Source data are available online for this figure.

et al, 2020). The interface comprises seven beta-sheets of each protomer shielded by several alpha helices.

We identified the key conserved residues for nucleotide base recognition: glutamate (E419 in RGC1; E379 in NeoR) and cysteine (C488 in RGC1; C448 in NeoR) (Fig. 4A). Substituting these residues with lysine and aspartate was previously shown to convert the homodimeric fungal CaRGC (from Catenaria anguillulae, closely related to BeRGC) into an adenylyl cyclase (AC) (Scheib et al, 2018). Additionally, two aspartates (D379/D423 in RGC1; D339/D383 in NeoR) are critical for divalent metal ion binding, which facilitates nucleotide interaction by stabilizing the negatively charged phosphate groups. Notably, the base recognizing residues and the metal ion binding aspartates reside on different protomers.

Given that all essential residues for GTP binding are conserved, we conducted enzymatic studies to elucidate the functional mechanism of the NeoR/RGC1 heterodimer. Both full-length proteins were co-expressed in HEK293T cells. Green-light of

522 nm applied to crude membrane fractions, which induced cGMP production and was quantified by reverse-phase HPLC. Cyclase activity peaked at a physiological pH of 7.4–8.0 and magnesium supported the highest enzymatic activity without dark activity, a condition utilized in further experiments (Fig. 4B). We determined the Michaelis-Menten constant ($K_M$) for heterodimeric RGC1/NeoR to be $0.55 \pm 0.03$ mM, which is comparable to that of homodimeric CaRGC ($0.47 \pm 0.05$ mM) (Fig. 4C). Due to the unknown heterodimer concentration in the sample, the value of $k_{cat}$ cannot be revealed from our data. Pursuing the functional impact of the red photoreceptor NeoR on the enzymatic parameters of the heterodimers, we switched NeoR₆₉₀ to NeoR₃₆₇ and measured kinetics. While the substrate affinity as indicated by the $K_M$ remained mostly unaffected, the maximal catalytic turnover showed some decrease upon photoconversion to NeoR₃₆₇ by illumination with far red light (Fig. EV2). However, we did not observe a significant change in activity upon photoswitching of the NeoR subunit under saturated GTP concentrations (2 mM).

## A functional asymmetry in substrate binding sites of heterodimeric RGCs

To investigate the role of each pseudo-symmetric active site, we modified the substrate specificity of RGC1 and NeoR towards ATP (AC) through mutagenesis. If both sites were catalytically active, modification of individual subunits should have resulted in heterodimeric enzymes capable of producing both cGMP and cAMP.

However, when we combined RGC1-WT with NeoR-AC (E379K/C448D), substrate specificity remained unchanged, but cGMP production in the light nearly doubled, and some minor dark activity appeared (~10%) (Fig. 4D). In contrast, pairing RGC1-AC (E419K/C488D) with NeoR-WT shifted substrate specificity towards ATP, completely abolishing cGMP production while preserving the overall turnover. When both AC variants were combined, light-triggered cAMP synthesis increased twofold accompanied with dark activity, reflecting the additive effect of both modified subunits. The lack of cAMP production observed for NeoR-AC indicates that only one of the two pseudo-symmetric NTP binding sites is catalytically active. In this active site, depicted in Fig. 4A, RGC1 contributes the base-discriminating residues (E419/C488) defining substrate specificity towards GTP, while the NeoR subunit provides the metal-binding aspartates (D339/D383), needed for nucleotide binding.

To further investigate the functional role of the second non-catalytic site, we disrupted the metal binding aspartate in RGC1 (RGC1-D379A) (Fig. 4D). This mutation prevents metal and thus nucleotide binding to the non-catalytic site, regardless of the chemical nature of the respective base. We observed a striking ninefold increase in light-triggered activity compared to the wild-type heterodimer. We hypothesize that in the WT protein, GTP binds to the non-catalytic site resulting in a partial allosteric inhibition of the catalytic center, that lowers the activity. Such an allosteric inhibition would be released by the RGC1-D379A mutation, since GTP binding is abolished. We utilized the RGC1-AC/NeoR-WT variant to disentangle nucleotide turnover from a putative allosteric regulation, but neither GTP nor cGMP affected cAMP production, even after NeoR photoconversion (Fig. EV2C).

## Enhanced substrate affinity in the soluble cyclase heterodimers

To investigate the impact of heterodimerization on the catalytic function at the level of the isolated cyclase, we expressed and purified the water-soluble cyclase core domains of all *R. globosum* RGCs, specifically RGC1-cat, RGC2-cat, and NeoR-cat. While RGC1-cat exhibited robust cGMP-synthesis indicating an active homodimer, no cGMP was produced from RGC2-cat despite 88% sequence identity between both cyclases. NeoR-cat also lacked cyclase activity. When RGC1-cat was mixed with equimolar NeoR-cat, $K_M$ was reduced from 5.53 mM obtained for homodimeric RGC1-cat alone to 1.54 mM in the RGC1-cat/NeoR-cat mixture. This accounts for the building of catalytic active heterodimers with increased substrate affinity, although some cGMP likely arises from homodimeric RGC1-cat. The maximal substrate turnover ($V_{max}$) was reduced to 50% (0.49 vs 0.96 µmol*min$^{-1}$*mg$^{-1}$) of the value observed for homodimeric RGC1-cat, which may account for the presence of only one catalytic active site in the heterodimeric

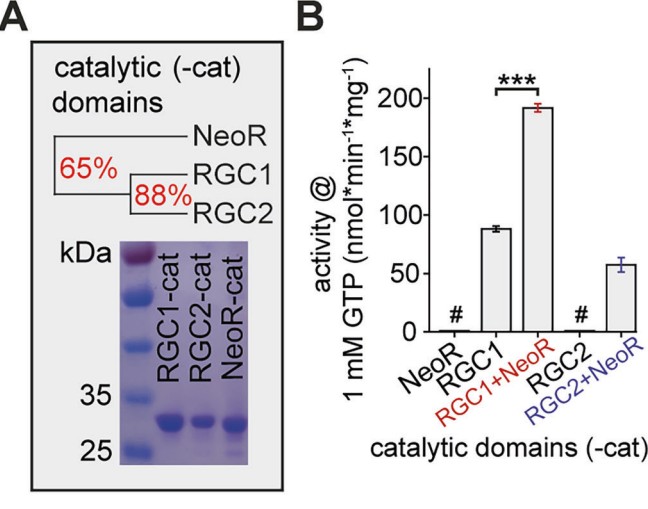

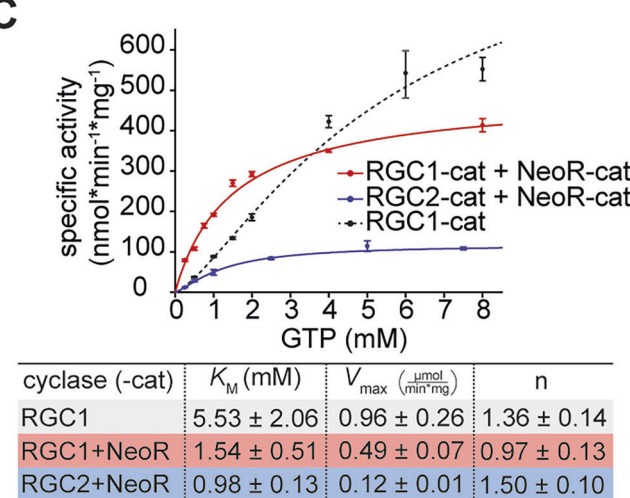

**Figure 5. Functional assays with soluble catalytic domains.**

(A) Clustal-based sequence identity tree of NeoR-cat, RGC1-cat, and RGC2-cat. Coomassie-stained SDS-page of His-tag purified catalytic domains. (B) Guanylyl cyclase activity of soluble catalytic domains at 1 mM GTP/2 mM MnCl$_2$, detected by reverse-phase HPLC. $n = 3$ (technical replicates), mean ± S.E., # = not detectable, unpaired t-test: ***$p = 2 \times 10^{-5}$. (C) Enzyme activity (Hill fits) of RGC1-cat and equimolar mixtures as indicated. The molar ratio of GTP and MnCl$_2$ is 1:2. The total amount of protein was kept constant. mean ± S.E., $n = 3$ (technical replicates). Source data are available online for this figure.

complex (Fig. 5C). Notably, also RGC2-cat/NeoR-cat form catalytic active heterodimers with a $K_M$ of 0.98 mM, despite both individual domains being inactive.

The functional asymmetry of the pseudo-symmetric active sites is also evident in RGC1-cat and NeoR-cat (Fig. EV3), presenting it as an inherent property of the heterodimeric cyclase core. Thus, this asymmetry likely applies to the native protein complex in the fungus as well.

The lack of catalytic activity observed for NeoR-cat may be explained by the absence of a conserved NxxxR motif (T455 and K459 in NeoR) (Fig. 4A), for which the arginine residue is discussed to be crucial for stabilizing the transition state during the cyclization reaction (Yan et al, 1997; Steegborn, 2014). This motif is

highly conserved in the active site of class III cyclases and is present in both RGC1 and RGC2 from *R. globosum*. In heterodimeric systems with a single catalytic active site, such as mammalian soluble adenylyl cyclase (sAC) or transmembrane adenylyl cyclase (tmAC), this motif is absent—just as in NeoR-cat (Fig. 4A, alignment). Nevertheless, introducing T455N and K459R mutations in NeoR-cat to mimic the active sites of RGC1 did not restore the catalytic activity (Fig. EV3).

## Discussion

Phototaxis of zoospores from chytrid fungi is a long-known process, and the involvement of a microbody-lipid globule complex (MLC) in light perception has been discussed for several decades (Chambers et al, 1967; Kazama, 1972; Barr and Hartmann, 1976; Powell, 1976). Although details of the molecular mechanisms are largely unknown, it recently became clear that the light-sensing machinery of these species possess heterodimeric rhodopsin guanylyl cyclases (RGCs) (Broser et al, 2023).

Our functional studies revealed that these proteins have only one catalytically active GTP-binding site, while the second degenerated site appears to exert an allosteric regulation. As with heterodimeric class III cyclases in mammals, this functional asymmetry provides enhanced regulatory control. The over nine-fold increase in activity resulting from a single point mutation that abolishes nucleotide binding suggests a regulatory role for GTP-binding in the regulation of heterodimeric RGCs, but this is yet to be experimentally confirmed. Nevertheless, it indicates that the accessible dynamic range of the complex is not fully covered by the light-activation of its RGC1 subunit. Therefore, it is tempting to suggest a functional role of the NeoR chromophore in the allosteric control of cGMP production, as previously proposed (Broser et al, 2020). However, we observed no significant effect of NeoR photoconversion regarding the activity of recombinantly expressed NeoR/RGC1 in human cells (HEK293T). Similar to mammalian heterodimeric class III cyclases, which are tightly regulated by various chemical stimuli such as bicarbonate, forskolin, calmodulin, G-proteins, and heme (Guo et al, 2009; Steegborn, 2014), fungal heterodimeric RGCs may require additional factors absent in the expression host, thus limiting their modulation in this system.

Our transcriptome analysis indicates that the mRNA level for NeoR in zoospores is roughly equivalent to the combined amount of RGC1 and RGC2, thus matching the ratio needed to form functional heterodimers. This suggests a highly regulated expression control already at the level of transcription. Given that RGC1 or 2 are only functional in conjunction with NeoR, we propose that NeoR associates with either RGC1 or 2 by forming heterodimeric complexes in fungal zoospores. Supporting this, NeoR strongly colocalizes with RGC1 or 2 when recombinantly expressed in mammalian cells (Broser et al, 2020). The low abundance of RGC2 mRNA compared to RGC1 is in line with our observation that despite being arranged in tandem, both genes are transcribed individually. It further indicates that the spectral sensitivity of the zoospores is dominated by RGC1 rather than RGC2. Nevertheless, detecting phototaxis of *R. globosum* zoospores has proved challenging. Unlike motile algae, chytrids are known to colonize a wide range of diverse habitats, acting as saprobionts in soil and lakes or infecting various eukaryotes, such as algae and amphibians

(Berger et al, 1998; Kobayashi et al, 2023). Thus, their movement and its directionality during the transient zoospore stage (lasting about 3 h under our laboratory conditions) may be influenced by other factors, such as chemotactic signals, which could dominate over light in the given setting (Moss et al, 2008).

Utilizing the intrinsic fluorescence of the NeoR subunit, we localized heterodimeric RGC photoreceptors adjacent to the lipid droplet within a chytrid-specific tubular structure, known as the rumposome (Barr and Hartmann, 1976; Laundon et al, 2022), which is part of the microbody-lipid globule complex (MLC). To our knowledge, this is the first example of a native retinal photoreceptor being directly visualized in its living host, without the need for external labeling or genetic modification. However, our approach does not distinguish between NeoR/RGC1 and NeoR/RGC2 complexes and, therefore, cannot resolve the spatial distribution of these two heterodimers, which are sensitive to green and blue light, respectively (Broser et al, 2020).

MLCs of fungal zoospores have been categorized based on morphological differences (Powell, 1978; Galindo et al, 2024). The MLC of *R. globosum* is of type 1B$_2$ and characterized by a single dominant lipid droplet in conjunction with the RGC-housing rumposome. In contrast, Type 4A MLCs are present in *B. emersonii*, containing multiple smaller lipid droplets associated with a single cisterna, termed backing membrane, where *Be*RGC was localized by antibody staining (Avelar et al, 2014).

Given that homodimeric *Be*RGCs and heterodimeric RGCs function similarly and are both localized in eyespot-like organelles near the flagellum, it is reasonable to conclude that heterodimeric RGCs play a comparable role in chytrid phototaxis.

Notably, NeoR's bistability enabled single-molecule localization microscopy (SMLM) with a mean localization precision of 32 nm, revealing a periodic spatial pattern in NeoR fluorescence. It is known that proton-pumping microbial rhodopsins arrange themselves in 2D lattices when densely packed in the membrane (Henderson et al, 1990; Müller et al, 2000). Yet, the 30 nm spacing observed in SMLM is too large to correspond to packed and distributed dimeric rhodopsin complexes. Instead, we attribute this pattern to the embedding of the photosensors in the hexagonal arranged membrane tubes that constitute the rumposome. Furthermore, SMLM allowed us to determine the number of ~3500 NeoR molecules per zoospore, though this value may be slightly underestimated due to photobleaching. Considering our transcriptomic analysis, we propose that the NeoR partners, RGC1 and RGC2, are expressed at equivalent levels in total.

The number of functional heterodimers per zoospores is substantially lower than the estimated channelrhodopsins in *C. reinhardtii* (≥10,000 per cell), albeit its eyespot is roughly the same size as the MLC of *R. globosum* (Beckmann and Hegemann, 1991; Harz et al, 1992). Unlike channelrhodopsins, which directly depolarize membranes upon activation in *C. reinhardtii*, RGCs generate the second messenger cGMP, leading to the opening of cGMP-gated ion channels, similar to what have been suggested for *E. gracilis* (Iseki et al, 2002; Nagel et al, 2002; Nagel et al, 2003). This signaling cascade, akin to vertebrate vision, suggests an amplification mechanism that allows photosensing with fewer photoreceptor molecules but at the expense of reduced time resolution (Beckmann and Hegemann, 1991).

Our data highlight the remarkable diversity of rhodopsin-based light perception shaped by biological evolution and demonstrate

super-resolution microscopy on native fluorescent rhodopsin to uncover details of the light-sensing machinery in chytrid zoospores.

# Methods

**Reagents and tools table**

| Reagent/Resource | Reference or Source | Identifier or Catalog Number |
|---|---|---|
| **Experimental models** | | |
| HEK293-T (cell-line) | ECACC | Cat #12022001 |
| Rhizoclosmatium globosum JEL800 | Cunliffe Lab (Univ. of Plymouth) | |
| *E. coli* (ArcticExpress (DE3)) | Agilent Technologies, Santa Clara, CA, USA | Cat #230192 |
| *P. pastoris* (SMD1168) | Invitrogen | Cat # C17500 |
| **Recombinant DNA** | | |
| piCZ-alpha | Invitrogen | Cat #V19520 |
| pEGFP-C1 | Clontech, Takara Bio, Kusatsu, Shiga Prefecture, Japan | Discontinued by Clontech |
| pET21(+) | Novagen/Merck Group, Darmstadt, Germany | Cat # 69770 |
| **Antibodies** | | |
| Anti-1D4 (mouse) | Invitrogen | Cat # MA1-722 |
| Anti-mouse-HRP | Invitrogen | Cat # 32230 |
| **Oligonucleotides and other sequence-based reagents** | | |
| All genes and primers | IDT/ Danaher Corp, Coralville, USA | NeoR, RGC1, RGC2, CaRGC as used in (Scheib et al, 2018; Broser et al, 2020; Broser et al, 2023) |
| Primer for RT-PCR | This study | Appendix Table S2 |
| **Chemicals, Enzymes and other reagents** | | |
| StreptactinXT DY-649 conjugate | IBA Lifesciences, Göttingen, Germany | Cat # 2-1568-050 |
| All-*trans* retinal | Sigma/Merck Group, Darmstadt, Germany | Cat # R2500 |
| NileRed | ThermoFisher Scientific Inc., Waltham, USA | Cat # N1142 |
| DDM (n-Dodecyl β-maltoside) | Glycon, Luckenwalde, Germany | Cat # D97002-C |
| CHS (Cholesterylhemisuccinat) | Sigma/Merck Group, Darmstadt, Germany | Cat # C6512 |
| QIAGEN RNeasy Plant Mini Kit | Qiagen, Germany | Cat # 79404 |
| ReverTraAce cDNA synthesis kit | Toyobo, Japan | Cat # FSK-101 |

| Reagent/Resource | Reference or Source | Identifier or Catalog Number |
|---|---|---|
| Pfu polymerase | Agilent Technologies, Santa Clara, CA, USA | Cat # 600255 |
| TurboFect | ThermoFisher Scientific Inc., Waltham, USA | Cat # R0531 |
| cGMP | Jena Bioscience, Germany | Cat # NU-1501 |
| GTP | Jena Bioscience, Germany | Cat # NU-1047 |
| cAMP | Jena Bioscience, Germany | Cat # NU-1503 |
| ATP | Jena Bioscience, Germany | Cat # NU-1049 |
| **Software** | | |
| Adobe Illustrator | Adobe | 2024 |
| Pymol | Schrödinger | 2.6.0 |
| tximport 1.32.0 | Soneson et al, 2015 | |
| DESeq2 1.44.0 | Love et al, 2014 | |
| Origin 2018 | OriginLab, Narthhampton, MA, USA | 2018 |
| ClarityChrom | Knauer, Berlin, Germany | 8.7.0.107 |
| Fiji ImageJ | NIH, USA | 1.54f |
| Matlab | The MathWorks, Inc, Portola Valley, CA, USA | R2023a |
| SMAP | Ries, 2020 | |
| **Other** | | |
| Super-resolution microscopy | | |
| Axiovert 200 microscope | Zeiss, Germany | N/A |
| α–Plan-Apochromat 100×/1.46 | Zeiss, Germany | N/A |
| iXon DU 897 EMCCD camera | Andor Technology Ltd., Northern Ireland | N/A |
| Chromatography | | |
| HiPrep 26/10 DS | Cytiva, Marlborough, MA, USA | Cat # 17508701 |
| HiTrap FF crude | Cytiva, Marlborough, MA, USA | Cat # 11000458 |
| 4Flow Strep-Tactin XT | IBA Lifesciences, Göttingen, Germany | Cat # 2-5023-001 |
| Supelco C18 column | Sigma-Aldrich, St. Louis-MO, USA | |

## Preparation of *Rhizoclosmatium globosum* JEL800 zoospores

*Rhizoclosmatium globosum* JEL800 was cultivated on PmTG-agar plates (0.1% peptonized milk, 0.1% tryptone, 0.5% glucose, 1.5% agar-agar, and 50 µg/ml streptomycin) and kept at 4 °C. For experimental use, 5 ml PmTG suspension was inoculated and incubated at room temperature (~22 °C) in darkness for 3 days. Subsequently, 500 µl of the suspension was transferred onto a freshly prepared PmTG-agar plate and incubated under the same conditions.

To induce sporulation, 7 ml sterile double-distilled water was added to the plate, and after 30 min, the zoospore-containing water was filtered through a 10 µm and 5 µm cell sieve (pluriSelect, Leipzig, Saxony, Germany). Then, zoospores were processed according to specific experimental requirements (see additional method sections).

## Zoospore RNA-extraction and reverse-transcriptase PCR

Zoospores harvested from five petri dishes were centrifuged for 5 min at $4000 \times g$. The cell pellet was stored at $-80$ °C until RNA extraction. Total RNA was extracted by using QIAzol/chloroform reagent following the manufacturer's instructions QIAGEN RNeasy Plant Mini Kit (QIAGEN, Germany, Cat. No. 79404). Briefly, cells were disrupted in QIAzol reagent, passed through one-time chloroform extraction, and total RNA was extracted from the aqueous phase using RNeasy Mini kit. After quantification of total RNA amount via UV-absorption, reverse transcription and real-time PCR was done using a ReverTraAce cDNA synthesis kit (Toyobo, Japan) (Yamasaki et al, 2016). The primer listed in Appendix Table S2 were used to amplify the RT-PCR products.

## Fluorescence spectroscopy

Following sieving, spores were centrifuged and resuspended in double-distilled water to an optical density of $OD_{600} = 2$. Fluorescence spectra were immediately recorded using a Horiba Fluoro-oMax 4 spectrometer with FluoresEssence™ 2.5.2. For photobleaching, spores were illuminated in the cuvette with a 680 nm LED for 60 min, followed by fluorescence recovery using a 375 nm LED for 1 min.

## Confocal fluorescence microscopy of *R. globosum* and image analysis

Harvested spores were centrifuged at $1000 \times g$ for 10 min, the supernatant was removed, and cells were resuspended in 1 ml phosphate-buffered saline (PBS; 136 mM NaCl, 2.7 mM KCl, 10 mM $Na_2HPO_4$, 1.76 mM $KH_2PO_4$, pH 7.4). Salt concentrations exceeding 100 mM inhibit zoospore motility. To capture images of each life cycle stage (excluding zoospores), 5 ml of PmTG medium was inoculated with a single *R. globosum* colony and incubated for 4–5 days prior to imaging. This unsynchronized liquid culture contains individuals of various life cycle stages. Lipid staining was performed by incubating spores with 0.5 µg/ml NileRed for 1 h, followed by two PBS washes. The cells were then resuspended in PBS containing 0.25% low-melting agarose to prevent movement and transferred on a glass dish for microscopy. Imaging was performed using a spinning disc confocal microscope, composed of

an Olympus IX83 Inverted Microscope (Shinjuku, Japan) and an EMCCD Camera (Andor iXon Ultra 888) equipped with either a 100x or 150x oil immersion objective and 405 (5 mW), 561 (50 mW), and 640 nm (50 mW) lasers. Image analysis was conducted using ImageJ, with brightness adjustments and background subtraction performed on differential interference contrast (DIC) images using a blank sample. Fluorescence images were also background-subtracted. Individual zoospores were photobleached using a 640 nm laser, and the NeoR far-red state was subsequently recovered by exposure to 405 nm laser light. The intensity of the fluorescence was determined by integrating a $25 \times 25$ pixel region surrounding the fluorescent organelle using Fiji ImageJ.

## Super-resolution microscopy

Zoospores from a 10 cm Petri dish were harvested, sieved, and centrifuged at $4000 \times g$ for 10 min. The resulting cell pellet was resuspended in 100 µl PBS, and 20 µl of the suspension was transferred to a circular glass microscopy dish. The dish was placed in a 50 °C warming cabinet for 5 min, followed by a 2-min incubation at $-20$ °C. To immobilize the cells, 20 µl of 1% PBS-Agarose (pre-warmed to ~50 °C) was carefully added to the dish. PALM (Betzig et al, 2006) was performed using a custom-built single-molecule microscope equipped with 405-nm (100 mW) and 642-nm (200 mW) lasers (Coherent, Santa Clara, CA, USA). These lasers were coupled into an inverted Zeiss Axiovert 200 microscope, focusing on the back-focal plane of a high numerical aperture objective (α-Plan-Apochromat 100×/1.46; Zeiss, Germany) for highly inclined illumination (~2 W/cm²) (Tokunaga et al, 2008). Emission light, filtered for NeoR (Alexa 647), was captured using a back-illuminated iXon DU 897 EMCCD camera (Andor Technology Ltd., Northern Ireland) cooled to $-80$ °C. The 642-nm laser provided constant illumination, while the 405-nm activation laser power was gradually increased to maintain consistent localization rates per frame. Lateral drift was corrected using a custom written MATLAB analysis software (see below). No significant defocusing was observed during measurements, so further corrections were unnecessary.

## Localization analysis

Data was analyzed using the SMAP platform (Ries, 2020). Typically, 10,000 frames with an integration time of 35 ms were acquired for reconstruction and analysis, with lateral drift corrected via SMAP's built-in cross-correlation. NeoR localizations averaged 350 photons, yielding a mean localization precision of 32 nm according to the fit derived from photon statistics via above referenced SMAP platform. Localization data (Figs. 3 and EV1) was rendered using Thomson blurring, representing each localization as a 2D Gaussian with width based on precision, determined by photon counts and background (Thompson et al, 2002). All analyses were performed in MATLAB, using raw localization coordinates, with processed images used solely for visualization.

## Simulations of rumposome substructure and image analysis

Data analysis was performed using custom Python scripts. Intensity profiles along rumposome structure from Thompson plots were

processed with Fiji. The intensity versus rumposome length for eight structures (Fig. EV1C,D), after applying a rolling ball background correction, was autocorrelated using a custom Python script. The first peak of the autocorrelation plot (distance lag) was fitted, yielding an approximate spacing of 30 nm. A simulation of a hexagonal pattern with a 30 nm horizontal pitch, modeled as Gaussian distributions with a full width at half maximum (FWHM) of 30 nm, represented NeoR localization. This distribution was further convoluted with localization accuracy to simulate precision, along with the random distribution of NeoR within the membrane at the perimeter of the tubular structure. The simulated pattern, $200 \times 200$ distributions, was rotated in 5-degree increments between 0 and 360 degrees to simulate varying projections of hexagonal distributions onto the x–y plane. We then analyzed the intensity by examining the simulated NeoR positions along a potential rumposome (Fig. EV1D, middle inset). and applied a frequency analysis to the resulting periodic pattern. This revealed a central peak at ~30 nm, confirming that a hexagonal pattern with a 30 nm pitch, even under random orientations with respect to the measurement plane, would indeed result in an averaged auto-correlation peak at similar wavelength (Fig. EV1D, inset).

## Transcriptomic data processing

Previously published transcriptome data was obtained from the NCBI Sequence Read archive (SRA) under accession number SRP350860 (Laundon et al, 2022). Reads were mapped to the *Rhizoclosmatium globosum* JEL800 genome and EST sequences, downloaded from MycoCosm (https://mycocosm.jgi.doe.gov/Rhihy1/Rhihy1.home.html) (Mondo et al, 2017), using Salmon 1.10.3 (Patro et al, 2017). A decoy-aware index was constructed using both the reference transcriptome and genome. Salmon quantification was performed with the 'validateMappings' and 'gcBias' options. Transcript-level quantifications were imported for differential expression analysis using tximport 1.32.0 (Soneson et al, 2015). Differential gene expression analysis was conducted with DESeq2 1.44.0 in R (Love et al, 2014). Normalized count data were subsequently exported and visualized.

## Expression of heterodimeric RGCs in HEK293T cells

For expression in HEK293T cells, full-length rhodopsin guanylyl cyclases (RGC1, RGC2, NeoR) were cloned into the pEGFP-C1 vector (Clontech, Takara Bio, Kusatsu, Shiga Prefecture, Japan). The genes (RGC1/NeoR; RGC2/NeoR) were arranged in the following order: N-terminus - RGC1 or RGC2 + Strep-Tag II (WSHPQFEK) + T2A-site + NeoR + 1D4-Tag (TETSQVAPA) - C-Terminus. Site-directed mutagenesis was performed using the QuikChange protocol with *Pfu* polymerase (Agilent Technologies, Santa Clara, CA, USA). Expression was conducted as previously described with minor modifications (Broser et al, 2020). Briefly, HEK293T cells were cultured in 150 cm² cell culture flasks with Dulbecco's modified Eagle's medium (DMEM) supplemented with 10% fetal bovine serum (FBS) and Penicillin-Streptomycin (Gibco, Thermo Fisher Scientific). Cells were transfected using TurboFect or polyethylenimine (PEI), with 8 µM all-*trans* retinal. After two days, cells were harvested in Dulbecco's phosphate-buffered saline (DPBS; Gibco, Thermo Fisher Scientific), containing 1x cOmplete™ EDTA-free protease inhibitor cocktail (Roche,

Switzerland). Following a wash step, cell pellets were frozen in liquid nitrogen and stored at −80 °C for future use.

## *E. coli* expression and purification of soluble RGC domains

The catalytic domains of RGC1 (aa: 365–551), RGC2 (aa: 365–550), and NeoR (aa: 325–509) were cloned into the pET21(+) vector (Novagen/Merck Group, Darmstadt, Germany) with a C-terminal His-tag. Expression was carried out in ArcticExpress (DE3) competent cells (Agilent Technologies, Santa Clara, CA, USA). An 800 ml culture was inoculated, grown to an $OD_{600} = 0.5$ at 37 °C and 200 RPM, then cooled to 18 °C at 160 RPM. Expression was induced with 1 mM isopropyl-beta-D-thiogalactoside. Cells were harvested the next day, resuspended in high-salt buffer (50 mM HEPES, 500 mM NaCl, 1 mM PMSF, pH 7.4), lysed using a high-pressure homogenizer (HTU Digi-French-Press, G. Heinemann, Germany) and centrifuged at $15,000 \times g$ for 15 min. The supernatant was purified via nickel-affinity chromatography on a 5 ml HiTrap FF crude column (Cytiva, Marlborough, MA, USA). After washing with 15 column volumes of high-salt buffer containing 20 mM imidazole, the protein was eluted with 500 mM imidazole. The buffer was then exchanged to HEPES-buffered saline (100 mM NaCl, 50 mM HEPES, pH 7.4) using a HiPrep 26/10 desalting column (Cytiva, Marlborough, MA, USA). Proteins were concentrated to 5–10 mg/ml and stored at −80 °C for future experiments. Protein concentrations were determined using following extinction coefficients at 280 nm ($\varepsilon_{NeoR-cat} = 24{,}075$ 1/M*cm; $\varepsilon_{RGC1-cat} = 32{,}555$ 1/M*cm; $\varepsilon_{RGC2-cat} = 32{,}555$ 1/M*cm) and protein masses ($m_{NeoR-cat} = 24{,}987$ g/mol; $m_{RGC1-cat} = 24{,}997$ g/mol; $\varepsilon_{RGC2-cat} = 25{,}179$ g/mol).

## Cyclase activity assay and cyclic nucleotide detection

To measure cyclase activity in HEK293T cells, the cell pellets were resuspended in HBS (100 mM NaCl, 50 mM HEPES, pH 7.4) supplemented with divalent ions (specified in the figure description). Cells were lysed with 30 strokes in a 2 ml Dounce homogenizer. The crude membrane fraction was washed by centrifugation (5 min, $8000 \times g$, 4 °C) and resuspended in HBS containing divalent ions. RGC amount was normalized by measuring the intrinsic fluorescence of the C-terminally expressed NeoR in the T2A-construct. For each assay, GTP was added to triplicate samples in 350 µl total volume, with reactions conducted either under full-intensity illumination using a bottom-transparent 96-multiwell plate placed on top of an Adafruit $64 \times 64$ RGB LED matrix or in darkness. Reactions were stopped at 2–3 time points within the enzyme's linear activity range by transferring 100 µl of the reaction mix to 100 µl of 0.1 N HCl. Denatured proteins were pelleted by centrifugation (10 min, $21{,}000 \times g$, 4 °C). For nucleotide detection, 30 µl of supernatant was applied to a C18 column (Supelco, Sigma-Aldrich, St. Louis, MO, USA) connected to a reversed-phase HPLC system (AZURA 6.1L system, Knauer, Berlin, Germany) and equilibrated with 100 mM potassium phosphate (pH 5.9, 4 mM tetrabutylammonium iodide, and 10% (v/v) methanol) at 1.2 ml/min flow rate and 30 °C. Soluble catalytic domain activity was measured in triplicate samples containing 50 µg of protein each, with $Mn^{2+}$ as a divalent ion. HPLC retention times were ~5 min for GTP, 8 min for cGMP, 10 min for ATP, and 16 min for cAMP (retention times may vary after each equilibration). Chromatograms were analyzed with ClarityChrom

8.7.0.107 software (Knauer, Berlin, Germany). Enzyme turnover was calculated using a cGMP standard curve with 0.05, 0.1, 0.2, 0.5, 1, 2, and 5 mM concentration cGMP or by determining the ratio of the cGMP peak area to total nucleotide content. For each time point, the mean cNMP concentration was calculated, and turnover per minute (slope) was obtained by linear fitting cNMP over time using Origin 2018 software (OriginLab, Northampton, MA, USA). For enzyme kinetics, data points were fitted with the Hill equation.

### RGC expression in *P. pastoris* and purification for UV/vis spectroscopy

Genes encoding *Rhizoclosmatium globosum* RGC1 (UniProtKB A0A1Y2CSF8), RGC2-rhodopsin (UniProtKB A0A1Y2CSL9; aa 1–319), and NeoR-rhodopsin (UniProtKB A0A1Y2CSJ0; aa 1–279) with a C-terminal Strep-Tag II were inserted into the pPICZ plasmid (Thermo Fisher Scientific, Waltham, MA, USA). Competent *Pichia pastoris* cells were prepared, transformed, selected, and expressed according to the manufacturer's protocol (EasySelect Pichia Expression Kit, Invitrogen, Carlsbad, CA, USA) with minor modifications. Rhodopsin expression was induced by adding 2.5% methanol and 5 μM all-*trans* retinal to the culture. After 24 h, the culture was harvested and lysed using a high-pressure homogenizer (HTU Digi-French-Press, G. Heinemann, Germany) with the addition of bovine DNase I (Roche, Switzerland) and cOmplete™ EDTA-free protease inhibitor cocktail (Roche, Switzerland). Membranes were isolated through successive centrifugation steps, and the resulting membrane pellet was resuspended in HEPES-buffered saline (HBS; 50 mM HEPES, 100 mM NaCl, pH 7.4) with cOmplete™ EDTA-free protease inhibitor cocktail (Roche, Switzerland). Solubilization was induced by adding 2% DDM and 0.4% CHS, and proteins were purified using a 4Flow Strep-Tactin XT column (IBA Lifesciences GmbH, Göttingen, Germany). Following affinity chromatography, the buffer was exchanged to HBS containing 0.02% DDM and 0.004% CHS using a HiPrep 26/10 desalting column (Cytiva, Marlborough, MA, USA). Spectra were then recorded for purified protein in HBS with detergent using a Shimadzu UV-2000 spectrophotometer with UVprobe v2.34 software (Shimadzu Corporation, Kyoto, Japan).

### SDS-page and immunoblotting

HEK293T-cell lysates were centrifuged at $12,000 \times g$ for 5 min, and the resulting pellet was resuspended in 50 μl HBS supplemented with 0.02% DDM and 0.004% CHS. After a 15-min incubation, 50 μl of 2x SDS-sample buffer (125 mM Tris, pH 6.8, 4% w/v SDS, 20% glycerol) was added. The mixture was heated at 95 °C for 3 min, centrifuged at $12,000 \times g$ for 5 min, and the supernatant was loaded onto a precast 12% Mini-Protean TGX gel (Bio-Rad, CA, USA). Proteins were transferred to a PVDF membrane via semi-dry blotting.

The membrane was blocked with PBS-T buffer (137 mM NaCl, 27 mM KCl, 10 mM $Na_2HPO_4$, pH 7.4, 0.1% v/v Tween-20) containing 3% bovine serum albumin for 1 h. After three washes with PBS-T, the membrane was incubated with a rhodopsin monoclonal antibody (1D4, Invitrogen, MA, USA) diluted 1:4000 for 1 h at room temperature for NeoR-detection. Following three additional PBS-T washes, a goat anti-mouse IgG peroxidase-conjugated secondary antibody (Invitrogen, MA, USA) diluted

1:2000 was applied for 1 h. Subsequently, a StreptactinXT DY-649 conjugate (IBA Lifesciences GmbH, Göttingen, Germany) diluted 1:4000 was incubated overnight for RGC1-detection. The membrane was then washed twice with PBS-T and once with PBS before imaging. For simultaneous detection of both proteins, Clarity Western ECL Substrate (Bio-Rad, CA, USA) was added, and chemiluminescence was detected using a chemidoc imaging system (Bio-Rad, CA, USA). The red fluorescence of RGC1-strep labeled with DY649 fluorophore was detected using the system's DY649 fluorescence detection settings.

## Data availability

All data and resources used in this study are available upon request. Original R code used for transcriptome analysis is available on github (https://github.com/DNAborn/Rhihy1). Python code for rumposome modeling is available here (https://github.com/klotzsch-lab/Rumposome). Any additional information required to reanalyze the data reported in this paper is available from the lead contact upon request.

The source data of this paper are collected in the following database record: biostudies:S-SCDT-10_1038-S44318-025-00452-x.

## Peer review information

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

## Acknowledgements

We thank Ben Boehlke and Dr. Thomas Korte for technical support. Further, we thank Prof. Michael Cunliffe and Kimberly Bird for providing the fungus. MB, EK, and PH acknowledge funding by the German Research Foundation (DFG) (509731234 (MB), KL 3278/2 (EK), and 431609106 (PH)). PH is a Hertie Professor and is supported by the Hertie Foundation. We acknowledge support by the Open Access Publication Fund of Humboldt-Universität zu Berlin.

## Author contributions

**Wayne Busse**: Conceptualization; Data curation; Formal analysis; Validation; Investigation; Visualization; Methodology; Writing—original draft; Writing—review and editing. **Enrico Klotzsch**: Resources; Software; Formal analysis; Validation; Investigation; Visualization; Methodology. **Yousef Yari Kamrani**: Investigation; Methodology. **Natalie Wordtmann**: Investigation. **Simon Kelterborn**: Data curation; Software; Formal analysis; Methodology. **Peter Hegemann**: Funding acquisition; Project administration; Writing—review and editing. **Matthias Broser**: Conceptualization; Formal analysis; Supervision; Funding acquisition; Validation; Methodology; Writing—original draft; Project administration; Writing—review and editing.

Source data underlying figure panels in this paper may have individual authorship assigned. Where available, figure panel/source data authorship is listed in the following database record: biostudies:S-SCDT-10_1038-S44318-025-00452-x.

## Funding

## Disclosure and competing interests statement

The authors declare no competing interests.

# Expanded View Figures

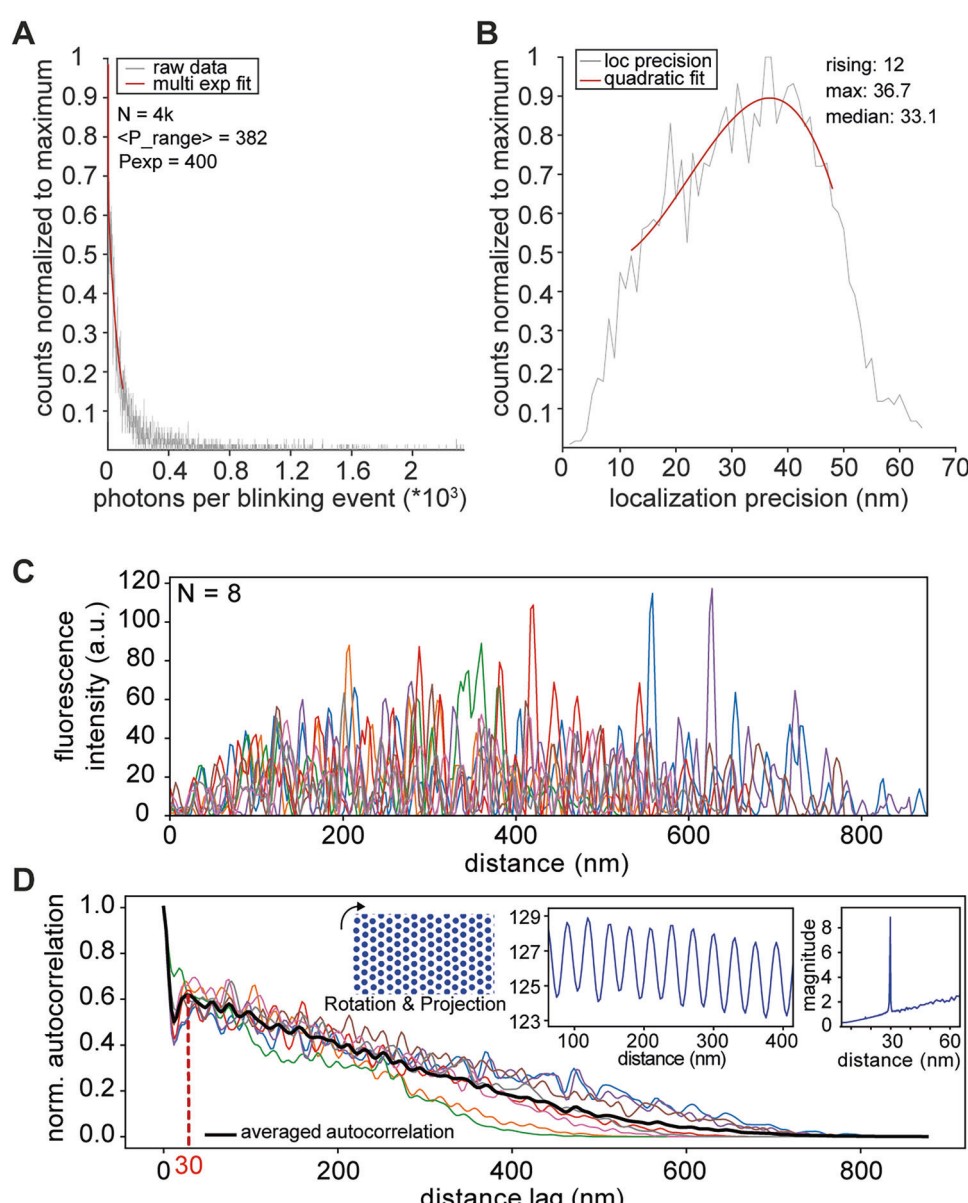

**Figure EV1. Single-molecule localization microscopy data analysis, related to Fig. 3.**

(A) Representative photon statistics for one individual zoospore. The SMAP software (Ries, 2020) was used to analyze blinking events recorded with the single-molecule localization microscope. The fits of the single blinking events resulted in x and y positions as well as background, photon number and localization precision. An exponential fit is applied to the distribution of photons per blinking event to determine the average photons per event. (B) Localization precision was calculated based on the photon count distribution, using the approach described in Ries (Ries, 2020) implemented in the SMAP software package. The localization precision was used to filter localizations with 50 nm and better, which serves as a threshold to exclude autofluorescence. The quadratic fit in (B) shows the estimated localization precision, with a median of 33.1 nm. (C) Background-corrected intensity profiles along the curved rumposome distributions highlight structured fluorescence patterns. (D) Autocorrelation functions from data in (C) are plotted over distance to assess periodicity in NeoR localizations, with the first peak detected at 30 nm (black line: averaged autocorrelation). The inset illustrates a simulated hexagonal distribution of NeoR localizations within the rumposome, based on experimentally determined spacing. The left panel shows the raw NeoR localization pattern, the middle panel presents the projection of the rumposome onto the x–y plane after rotation, and the right panel displays frequency analysis, confirming a periodic spatial pattern with a dominant peak at 30 nm. Localisation data were processed using the SMAP platform, ensuring robust drift correction and precision estimation.

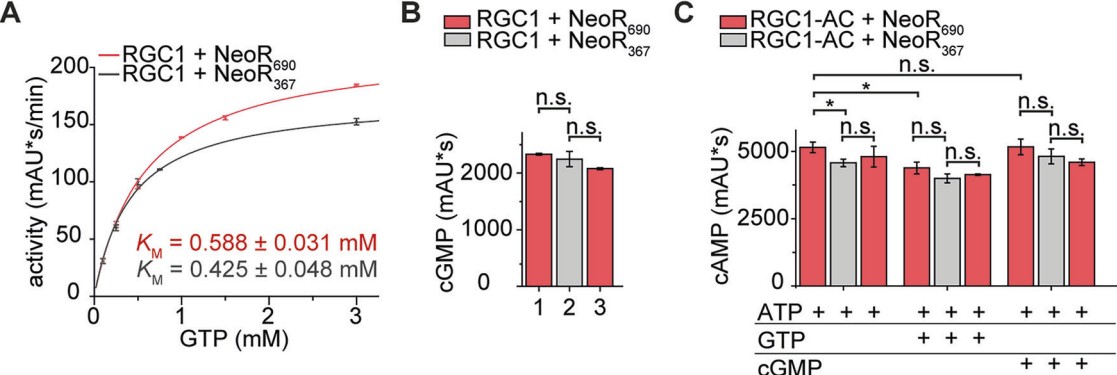

**Figure EV2. Functional assays with photoswitched NeoR, related to Fig. 4.**

(A) Michaelis-Menten kinetics of co-expressed RGC1/NeoR in HEK293T cells before and after photoconversion of NeoR, measured with 10 mM MgCl$_2$ and detected via reverse-phase HPLC. $n = 3$ (technical replicates), mean ± S.E. (B) End-point activity assay (light activation at 522 nm, 2 mM GTP/10 mM MgCl$_2$) with subsequent photoconversion of the NeoR subunit, 1: without pre-illumination, 2: photoconverted with 680 nm (20 min), 3: back-converted with 375 nm (1 min). $n = 3$ (technical replicates), mean ± S.D., unpaired t-test left-to-right: $p = 0.39$ (n.s. = not significant), $p = 0.12$ (n.s. = not significant). (C) End-point cyclase activity assay of co-expressed RGC1-AC/NeoR with 2 mM ATP/5 mM MgCl$_2$ and optionally cGMP or GTP (1 mM) added. NeoR was reversibly photoconverted using far-red (680 nm) and UV-light (375) as in (B). A minor decrease upon addition of GTP likely reflects competition with substrate ATP (Ruiz-Stewart et al, 2003). $n = 3$ (technical replicates), mean ± S.D., unpaired t-test left-to-right: *$p = 0.01$, *$p = 0.01$, $p > 0.05$ (n.s. = not significant, values are available in source data). Source data are available online for this figure.

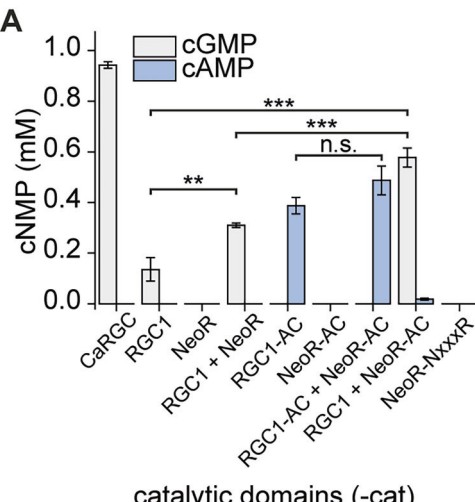

**Figure EV3. Activity assays of soluble catalytic domain variants, related to Fig. 5.**

(A) Endpoint enzyme assays (at 1 mM NTP/ 2 mM MnCl$_2$) with equimolar amounts of purified RGC-WT (*R. globosum*) catalytic domains and mutants, compared to homodimeric *Ca*RGC-cat (*C. anguillulae*). $n = 3$ (technical replicates), mean ± S.D., unpaired t-test left-to-right: **$p = 3 \times 10^{-3}$, ***$p = 2 \times 10^{-4}$, ***$p = 3 \times 10^{-3}$, $p = 0.06$ (n.s. = not significant). Source data are available online for this figure.

