## [Peer Review File · The EMBO Journal]

Super-Resolution Imaging of Native Fluorescent Photoreceptors in Chytrid Fungal Eyes

Wayne Busse, Enrico Klotzsch, Yousef Kamrani, Natalie Wordtman, Simon Kelterborn, Peter Hegemann, and Matthias Broser

Corresponding author(s): Matthias Broser (matthias.broser@hu-berlin.de)

Review Timeline:

Submission Date:	21st Jan 25
Editorial Decision:	26th Feb 25
Revision Received:	29th Mar 25
Editorial Decision:	11th Apr 25
Revision Received:	17th Apr 25
Accepted:	23rd Apr 25

Editor: Hartmut Vodermaier

Transaction Report:

Dr. Matthias Jan Broser
Humboldt Universität zu Berlin
Experimental Biophysics
Invalidenstrasse 42
Berlin, Berlin 10115
Germany

26th Feb 2025

Re: EMBOJ-2025-120252
Super-Resolution Imaging of Native Fluorescent Photoreceptors in Chytrid Fungal Eyes

Dear Dr. Broser,

Thank you for submitting your study on characterization and imaging of fungal NeoR-RGC to The EMBO Journal. It has now been assessed by three referees with expertise in zoospore vision, microbial rhodopsins, and super-resolution imaging, respectively, whose comments are copied below. All reviewers appreciate the interest and potential importance of the work, and would in principle support its publication. Nevertheless, they raise a number of specific concerns that would need to be addressed prior to acceptance. As you will see, most of these points refer to presentational aspects, better descriptions and quantifications; but especially referee 3 also has a few technical issues and clarification requests that may require additional experimentation.

Should you be able to adequately address these points, we would be happy to consider a revised version further for The EMBO Journal. Please note that it is our policy to consider only a single round of major revision, making it important to fully respond to all comments at the time of resubmission; therefore, please do not hesitate to get back to me in case you would like to clarify/discuss any of the referees' points or plans for addressing them ahead of time. We would also be open to extending the revision deadline if that should be helpful.

Detailed information on preparing, formatting and uploading a revised manuscript can be found below and in our Guide to Authors, and adhering to them as closely as possible shall greatly facilitate editorial processing upon resubmission. Thank you again for the opportunity to consider this work for The EMBO Journal, and I look forward to your revision in due time.

With kind regards,

Hartmut Vodermaier

9) To facilitate reproducibility and cross-laboratory adoption of methodologies, please structure the Materials & Methods section as outlined in our guide to authors, including a completed Reagents and Tools Table that can be downloaded from our author guidelines as well (<https://www.embopress.org/page/journal/14602075/authorguide#structuredmethods>).

10) Digital image enhancement is acceptable practice, as long as it accurately represents the original data and conforms to community standards. If a figure has been subjected to significant electronic manipulation, this must be clearly noted in the figure legend and/or the 'Materials and Methods' section. The editors reserve the right to request original versions of figures and the original images that were used to assemble the figure. Finally, we generally encourage uploading of numerical as well as gel/blot image source data; for details see: embopress.org/page/journal/14602075/authorguide#sourcedata

At EMBO Press, we ask authors to provide source data for the main manuscript figures. Our source data coordinator will contact you to discuss which figure panels we would need source data for and will also provide you with helpful tips on how to upload and organize the files.

In the interest of ensuring the conceptual advance provided by the work, we recommend submitting a revision within 3 months (27th May 2025). Please discuss the revision progress ahead of this time with the editor if you require more time to complete the revisions. Use the link below to submit your revision:

Link Not Available

Referee #1:

In this manuscript Busse et al. characterize heterodimeric RGCs proteins (Rhodopsin Guanylyl Cyclases only found in the light-sensing lipid-rich organelle of zoospores (MLCs) in some species of chytrid fungi) after discovering the autofluorescence capabilities of the NeoR protein, a type of RGC. By using the chytrid *Rhizoclosmatium globosum* the authors show how the photoswitching of its NeoR protein enables super-resolution microscopy, thus, allowing them to detect and quantify the NeoR proteins within individual zoospores and determine its localization within the rumposome (a chytrid-specific organelle). As expected, fluorescence tracking + transcriptomic analysis on different life-cycle stages confirm that RGCs are present mainly in the zoospore stage of *R. globosum* and that its expression level are congruent to the amount of RGCs that they have quantified with microscopy. Lastly, the authors performed functional assays to investigate the mechanism of activation of RGC heterodimers.

This study made by Busse et al. shows for the first time that the RGC heterodimers in zoospores of *R. globosum* (from the phylum Chytridiomycota) are located within their eyespot-like lipid-rich organelle (this was already confirmed for homodimeric RGCs in the zoospores of *Blastocladiella emersonii* from the phylum Blastocladiomycota). Thus, providing further evidence to the decades-long discussion on whether these structures are involved in chytrid phototaxis. Additionally, this is the first piece of evidence confirming that light-perception on these zoospores is mediated by heterodimeric RGCs occurring specifically within

the MLC's rumposome. This is an outstanding and beautifully crafted piece of work, with a very solid methodology, that contributes to the retinal protein biology and wider microbiology communities with several valuable assets, including: (1) a new approach to image NeoR proteins by using its autofluorescence; (2) provides new insights into the understudied MLC organelle of chytrid fungi, and remarkably about the mysterious Chytridiomycota-exclusive structure known as the rumposome; (3) characterizes the first naturally occurring photosystem that can be measured *in vivo*, and provide structural detail below conventional resolution limits; (4) sheds light into the underlying functional mechanics of these unique rhodopsins, including its activation and regulation. I very much enjoyed reading the manuscript and I consider it to be of high interest to the target audience of The EMBO journal and almost publishable as it is. Thus, I recommend its acceptance and publication after addressing my only main and minor comment:

1- Please use the term Microbody-Lipid globule Complex (MLCs) throughout the entire manuscript to describe the lipid-rich eyespot-like organelle of chytrid fungi. This term was established by Martha Powell in 1976 (<https://doi.org/10.1007/BF01279325>), a pioneer within the field that later created a very detailed classification of the different types of MLCs based on the presence/absence of different cytological components (10.1016/0303-2647(78)90038-2), including for example, the rumposome, and amount of lipid droplets (the authors need to cite both of Powell's studies). On this note, the authors should use Powell's MLC classification (10.1016/0303-2647(78)90038-2) to state that *R. globosum* zoospores (since they have a rumposome) have a MLC type 1B2 (also see: <https://doi.org/10.1016/j.cub.2024.08.016>). *R. globosum*'s MLC classification needs to be mentioned since future studies on the localization of other RGCs will heavily rely on the cytological architecture variations of these MLCs. This would be important, for example, for MLCs that lack a rumposome and thus, their RGCs may have a different localization (e.g., the type 1A of the zoospore of *Synchytrium microbalum* or the MLCs of type 4 found in the zoospores of *Blastocladiomycota*). Powell's classification is a good way to unify terminology within the community. Thus, the term lipid-rumposome-microbody (LRM) (coined posteriorly by Laundon et al.) should be used with caution and only after prioritizing Powell's classification (since Laundon et al only talks about one example of one type of MLC, the 1B2 type).

Minor comment:

- I look forward to the possibility of confocal immunofluorescent imaging of the NeoR autofluorescence coupled with RGC1/2 specific antibodies. It would be very useful to test if these heterodimer combinations are present in particular areas or patches within the rumposome or they are mixed indifferently along its surface. Could the authors comment on the discussion whether they expect RGC1/NeoR and RGC2/NeoR to have specific localizations given their functional differences?

Referee #2:

NeoR is a unique rhodopsin that the authors first discovered in 2020, and has the following extremely unique characteristics: 1. It is a bi-stable molecule that absorbs at 690 nm, close to near-infrared light, and has a second absorption at 350 nm. 2. It forms a heterodimer with RGC1 or RGC2, which are visible light absorbing molecules. All microbial rhodopsins form 2-10 homomultimers, but NeoR is the first example of a heterocomplex. RGCs and NeoR are found in chytrids, and are predicted to be involved in phototaxis in native cells, producing cGMP in response to light signals and thus regulating the activity of the CNG channel. In this manuscript, the authors first confirmed the gene expression of NeoR, RGC1, and RGC2 mainly at the zoospores stage in the fungus *Rhizoclosmatium globosum*, the original host of NeoR, and further revealed that it is localized in a domain called the rumposome by fluorescent imaging of the NeoR molecule directly. Furthermore, single molecule analysis showed that NeoR is distributed in a honeycomb structure with a spacing of 39 nm. This is, as the authors mentioned, the first naturally occurring photosystem that can be measured *in vivo*, with fluorescence microscopy providing structural detail below conventional resolution limits. Therefore, the above results are extremely valuable. In addition, the authors performed biochemical experiments to reveal the molecular functions of NeoR and RGCs. There was no significant effect on the enzyme activity of light at the absorption wavelengths of NeoR, 690 nm and 367 nm. Thus, the physiological significance of near-infrared and UV light remains unknown. On the other hand, it is very interesting that mutant experiments led to the hypothesis that allosteric activity inhibition occurs by GTP binding to the inactive site.

As mentioned above, this manuscript provides new insight into the characteristic expression and distribution pattern of NeoR in the host organism, and the unique molecular mechanism of NeoR and RGC. Thus, the manuscript is worthy of publication in EMBO.

Comment

Fig S1 shows a honeycomb structure of fluorescent signals spaced at 30 nm intervals. Is this 30 nm appropriate when dimeric rhodopsins are densely packed and distributed? For example, bacteriorhodopsin forms a trimeric complex and distributes densely to form a purple membrane (see the paper below). Please discuss by citing the paper.

D J Müller, J B Heymann, F Oesterhelt, C Möller, H Gaub, G Büldt, A Engel
Atomic force microscopy of native purple membrane
Müller et al. *Biochimica et Biophysica Acta (BBA) Bioenergetics* 1460(1), 27-38, 2000

Minore comment

Page 6 line 2. reference "Broser et al, 2012"

Referee #3:

In the manuscript "Super-Resolution Imaging of Native Fluorescent Photoreceptors in Chytrid Fungal Eyes Busse et al.", the authors exploit the intrinsic fluorescent characteristics of neorhodopsin to investigate the localization of rhodopsin guanylyl cyclases (RGC) in zoospores (and other stages in the life cycle) of the chytrid fungus *Rhizoclosmatium globosum* by single-molecule localization microscopy with a resolution of 30-35 nm. Furthermore, the authors include functional assays of recombinantly expressed (RGCs) heterodimers. Site-directed mutagenesis allowed them to generate a mutant with up to nine-fold increased cyclase activity.

Overall, the manuscript is of interest for a broad audience as it combines aspects from different disciplines like microscopy, mycology, and protein biochemistry. However, at the same time due to different topics the storyline of the manuscript needs significant improvement. Furthermore, sometimes methods, or data are not shown in a way that it would allow for reproduce these experiments in other labs due to the lack of information.

- specific major concerns essential to be addressed to support the conclusions

In the first part of the manuscript the authors show that they can use the SMLM for the analysis of NeoR distribution in Zoospores. They conclude, that due to the fact that NeoR builds dimers with the cyclase entity the presence of NeoR means also presence of cyclase. However, this is an assumption that needs either support by a suitable experiment or a clearly elaborated literature-based argumentation.

Using NeoR for SMLM is quite promising. However, would not it be possible to use the same technique with HEK cells heterologous expressing the cyclases? I would be very interesting to see where the proteins were located in HEK cells. And it would be a good link / possible argument to address a weak point of the manuscript. There are two general problems that should be solved:

The authors assume that the observation of Neorhodopsin means also the presence of RGC? However, this should be experimentally supported or at least clearly elaborated.

A second problem is the functional assay - the authors should convince the reader, that the results obtained with protein expressed in mammalian cells can be transferred to the protein function in the fungus?

Page 7: The authors detect high fluctuation in the density of 3,417 {plus minus} 1,543 per zoospore (N = 23). The authors should discuss this. What is the biological reason for this high deviation?

Fig. 3E. The authors should take care about reproducibility. There is missing information: How were the different stages of the chytrid lifecycle obtained. A description of the culture conditions, immobilization, and further important aspects should be incorporated in the Methods part.

"To our knowledge, this is the first naturally occurring photosystem that can be measured in vivo, with fluorescence microscopy providing structural detail well below conventional resolution limits." As this is a crucial finding of the manuscript, supporting data should be clearly elaborated. While Figure S1 shows the measurement of precision, the explanation needs improvement to allow reproduction. To improve in page 14/15 the following questions should be addressed: What was the acquisition time of each frame? What was the cut off to distinguish between NeoR signal and autofluorescence. How exactly was the localization precision defined? The mentioned custom python scripts should be made available via a public repository like eg. Github.

Overall statistical tests and values should be given to all figures presented.

Finally, the discussion requires revision, as not all results are addressed. Especially, results regarding single molecule microscopy are not considered. In addition, a conclusion or at least one or two conclusive sentences would be beneficial.

- minor concerns that should be addressed

In the introduction, the authors should also mention in the paper regarding phototaxis experiments with zoospores of *Allomyces reticulatus* (Saranak & Foster, 1997).

Page 3: The terminology of rhodopsin guanylyl cyclases is not consistent in the literature. Is there a reason why the abbreviation changed from RhoGC, via RhGC to RGC? Authors should try to use the abbreviation used before or to explain their choice.

Page 5. "Since chytrid zoospores are considered translationally inactive, the presence of RNA may result from inherited and inactive mRNA-loaded ribosomes (Medina et al, 2020)." This sentence needs rewriting if the authors want to highlight that the RNA is of maternal origin as transcription does not take place in zoospores.

"The lack of amplification products indicates that each gene is transcribed individually". As PCR tests with fungal gDNA tend to fail, I suggest the authors should use a more cautious way like "suggests", or "could be due to individual transcription of..."
„A1-retinal" Better use the expression all-trans-retinal which is quite more frequent in the rhodopsins field. Also "A2-(3-hydro-)retinal" should be adapted accordingly

Page 8. "[...] two putative cGMP-gated ion channels (Figure S1)". Figure S1 should be replaced with Figure S2

Figure 2 c. It would be nice enlarge this figure to see the described effects more clearly. If an additional figure is allowed, a single figure should be considered.

In figure 3Ciii the graphics is damaged and should be repaired

- any additional non-essential suggestions for improving the study (which will be at the author's/editor's discretion)

The paragraph "A functional asymmetry in substrate...". If the reader is not an expert working with these proteins, it is difficult to understand, why each respective mutation was produced. To avoid confusion the whole section needs further improvement

regarding explanation of the respective mutations etc.

Point-by-point response to the Reviewers comments:

Referee #1:

In this manuscript Busse et al. characterize heterodimeric RGCs proteins (Rhodopsin Guanylyl Cyclases only found in the light-sensing lipid-rich organelle of zoospores (MLCs) in some species of chytrid fungi) after discovering the autofluorescence capabilities of the NeoR protein, a type of RGC. By using the chytrid *Rhizoclostium globosum* the authors show how the photoswitching of its NeoR protein enables super-resolution microscopy, thus, allowing them to detect and quantify the NeoR proteins within individual zoospores and determine its localization within the rumposome (a chytrid-specific organelle). As expected, fluorescence tracking + transcriptomic analysis on different life-cycle stages confirm that RGCs are present mainly in the zoospore stage of *R. globosum* and that its expression level are congruent to the amount of RGCs that they have quantified with microscopy. Lastly, the authors performed functional assays to investigate the mechanism of activation of RGC heterodimers.

This study made by Busse et al. shows for the first time that the RGC heterodimers in zoospores of *R. globosum* (from the phylum Chytridiomycota) are located within their eyespot-like lipid-rich organelle (this was already confirmed for homodimeric RGCs in the zoospores of *Blastocladiella emersonii* from the phylum Blastocladiomycota). Thus, providing further evidence to the decades-long discussion on whether these structures are involved in chytrid phototaxis. Additionally, this is the first piece of evidence confirming that light-perception on these zoospores is mediated by heterodimeric RGCs occurring specifically within the MLC's rumposome. This is an outstanding and beautifully crafted piece of work, with a very solid methodology, that contributes to the retinal protein biology and wider microbiology communities with several valuable assets, including: (1) a new approach to image NeoR proteins by using its autofluorescence; (2) provides new insights into the understudied MLC organelle of chytrid fungi, and remarkably about the mysterious Chytridiomycota-exclusive structure known as the rumposome; (3) characterizes the first naturally occurring photosystem that can be measured in vivo, and provide structural detail below conventional resolution limits; (4) sheds light into the underlying functional mechanics of these unique rhodopsins, including its activation and regulation. I very much enjoyed reading the manuscript and I consider it to be of high interest to the target audience of The EMBO journal and almost publishable as it is. Thus, I recommend its acceptance and publication after addressing my only main and minor comment:

We thank Reviewer 1 for the very positive review of our manuscript.

1- Please use the term Microbody-Lipid globule Complex (MLCs) throughout the entire manuscript to describe the lipid-rich eyespot-like organelle of chytrid fungi. This term was established by Martha Powell in 1976 (<https://doi.org/10.1007/BF01279325>), a pioneer within the field that later created a very detailed classification of the different types of MLCs based on the presence/absence of different cytological components (10.1016/0303-2647(78)90038-2), including for example, the rumposome, and amount of lipid droplets (the authors need to cite both of Powell's studies). On this note, the authors should use Powell's MLC classification (10.1016/0303-2647(78)90038-2) to state that *R. globosum* zoospores (since they have a rumposome) have a MLC type 1B2 (also see: <https://doi.org/10.1016/j.cub.2024.08.016>). *R. globosum*'s MLC classification needs to be mentioned since future studies on the localization of other RGCs will heavily rely on the cytological architecture variations of these MLCs. This would be important, for example, for MLCs that lack a rumposome and thus, their RGCs may have a different localization (e.g.,

the type 1A of the zoospore of *Synchytrium microbalum* or the MLCs of type 4 found in the zoospores of *Blastocladiomycota*). Powell's classification is a good way to unify terminology within the community. Thus, the term lipid-rumposome-microbody (LRM) (coined posteriorly by Laundon et al.) should be used with caution and only after prioritizing Powell's classification (since Laundon et al only talks about one example of one type of MLC, the 1B2 type).

We are grateful for the valuable suggestions to which we totally agree. Especially for considering future studies regarding other chytrids, we have revised the manuscript accordingly by replacing the term LRM with MLC and have included the relevant references. Additionally, we have restructured the discussion part focusing on MLC architecture, specially addressing the roles of 1B2 and 4A type MLCs in chytrids.

Minor comment:

- I look forward to the possibility of confocal immunofluorescent imaging of the NeoR autofluorescence coupled with RGC1/2 specific antibodies. It would be very useful to test if these heterodimer combinations are present in particular areas or patches within the rumposome or they are mixed indifferently along its surface. Could the authors comment on the discussion whether they expect RGC1/NeoR and RGC2/NeoR to have specific localizations given their functional differences?

This is a very interesting point – however raising specific antibodies that discriminate between RGC1 and 2 is difficult. The antibody e.g. used by Avelar et al., *Curr. Biol.* 2014, detects the cytoplasmic cyclase domain of BeRGC. For RGC1 and 2 this domain is highly conserved (~88% identity) and therefore may not provide the specificity needed. In recombinant experiments in mammalian cells with fluorescence-tagged RGC1 and RGC2 (reported in Broser et al., *Nature Communication* 2020, see SI Fig. S4 in our response to Reviewer 3), the respective NeoR/RGC1 and NeoR/RGC2 heterodimers show distinct subcellular distributions, with NeoR/RGC1 forming large intracellular clusters, while NeoR/RGC2 are more distributed over the cell (see our response to Reviewer 3). However, the spatial distribution of RGC1/NeoR and RGC2/NeoR in the fungus cannot be resolved as we now point out on page 14:

“However, our approach does not distinguish between NeoR/RGC1 and NeoR/RGC2 and, therefore, cannot resolve the spatial distribution of these two heterodimers, which are sensitive to green and blue light, respectively.”

Referee #2:

NeoR is a unique rhodopsin that the authors first discovered in 2020, and has the following extremely unique characteristics: 1. It is a bi-stable molecule that absorbs at 690 nm, close to near-infrared light, and has a second absorption at 350 nm. 2. It forms a heterodimer with RGC1 or RGC2, which are visible light absorbing molecules. All microbial rhodopsins form 2-10 homomultimers, but NeoR is the first example of a heterocomplex. RGCs and NeoR are found in chytrids, and are predicted to be involved in phototaxis in native cells, producing cGMP in response to light signals and thus regulating the activity of the CNG channel. In this manuscript, the authors first confirmed the gene expression of NeoR, RGC1, and RGC2 mainly at the zoospores stage in the fungus *Rhizoclosmatium globosum*, the original host of

NeoR, and further revealed that it is localized in a domain called the rumposome by fluorescent imaging of the NeoR molecule directly. Furthermore, single molecule analysis showed that NeoR is distributed in a honeycomb structure with a spacing of 39 nm. This is, as the authors mentioned, the first naturally occurring photosystem that can be measured in vivo, with fluorescence microscopy providing structural detail below conventional resolution limits. Therefore, the above results are extremely valuable. In addition, the authors performed biochemical experiments to reveal the molecular functions of NeoR and RGCs. There was no significant effect on the enzyme activity of light at the absorption wavelengths of NeoR, 690 nm and 367 nm. Thus, the physiological significance of near-infrared and UV light remains unknown. On the other hand, it is very interesting that mutant experiments led to the hypothesis that allosteric activity inhibition occurs by GTP binding to the inactive site.

As mentioned above, this manuscript provides new insight into the characteristic expression and distribution pattern of NeoR in the host organism, and the unique molecular mechanism of NeoR and RGC. Thus, the manuscript is worthy of publication in EMBO.

We would like to thank Reviewer 2 for the very positive review of our manuscript.

Comment

Fig S1 shows a honeycomb structure of fluorescent signals spaced at 30 nm intervals. Is this 30 nm appropriate when dimeric rhodopsins are densely packed and distributed? For example, bacteriorhodopsin forms a trimeric complex and distributes densely to form a purple membrane (see the paper below). Please discuss by citing the paper.

D J Müller, J B Heymann, F Oesterhelt, C Möller, H Gaub, G Büldt, A Engel
Atomic force microscopy of native purple membrane
Müller et al. *Biochimica et Biophysica Acta (BBA) Bioenergetics* 1460(1), 27-38, 2000

Since the expected maximal dimensions of heterodimeric RGCs is ~7.5 nm, we do not think that the honeycomb structure with periodicity of 30 nm represents densely packed rhodopsins. Even hexameric proteorhodopsins, that form one of the largest rhodopsin oligomers, arrange in a hexagonal lattice with a spacing of only 8.8 nm (Klyszejko et al., 2007 *JMB* (<https://doi.org/10.1016/j.jmb.2007.11.030>)). We therefore propose that the 30 nm spacing observed in the fluorescence intensity pattern reflect rather the structure of the hexagonally arranged membrane tubes of the rumposome as determined by TEM. We now discuss this on page 15:

“It is known that proton-pumping microbial rhodopsins arrange themselves in 2D lattices when densely packed in the membrane (Müller et al, 2000; Henderson et al, 1990). Yet, the 30 nm spacing observed in SMLM is too large to correspond to packed and distributed dimeric rhodopsin complexes. Instead, we attribute this pattern to the embedding of the photosensors in the hexagonal arranged membrane tubes that constitute the rumposome.”

Minore comment

Page 6 line 2. reference "Broser et al, 2012"
This should be Broser et al, 2022

done

Referee #3:

In the manuscript "Super-Resolution Imaging of Native Fluorescent Photoreceptors in Chytrid Fungal Eyes Busse et al.", the authors exploit the intrinsic fluorescent characteristics of neorhodopsin to investigate the localization of rhodopsin guanylyl cyclases (RGC) in zoospores (and other stages in the life cycle) of the chytrid fungus *Rhizoclosmatium globosum* by single-molecule localization microscopy with a resolution of 30-35 nm. Furthermore, the authors include functional assays of recombinantly expressed (RGCs) heterodimers. Site-directed mutagenesis allowed them to generate a mutant with up to nine-fold increased cyclase activity.

Overall, the manuscript is of interest for a broad audience as it combines aspects from different disciplines like microscopy, mycology, and protein biochemistry.

We thank Reviewer 3 for the positive evaluation of the manuscript. Further we are grateful for the critical points that help to improve our manuscript (see below).

However, at the same time due to different topics the storyline of the manuscript needs significant improvement. Furthermore, sometimes methods, or data are not shown in a way that it would allow for reproduce these experiments in other labs due to the lack of information.

- specific major concerns essential to be addressed to support the conclusions

In the first part of the manuscript the authors show that they can use the SMLM for the analysis of NeoR distribution in Zoospores. They conclude, that due to the fact that NeoR builds dimers with the cyclase entity the presence of NeoR means also presence of cyclase. However, this is an assumption that needs either support by a suitable experiment or a clearly elaborated literature-based argumentation.

Using NeoR for SMLM is quite promising. However, would not it be possible to use the same technique with HEK cells heterologous expressing the cyclases? I would be very interesting to see where the proteins were located in HEK cells.

And it would be a good link / possible argument to address a weak point of the manuscript.

There are two general problems that should be solved:

The authors assume that the observation of Neorhodopsin means also the presence of RGC? However, this should be experimentally supported or at least clearly elaborated.

We are grateful to the reviewer for addressing this important point. We have already reported the recombinant expression of heterodimeric NeoR/RGC1 and NeoR/RGC2 complexes in mammalian cells (ND7/23 cells, see Broser et al., 2020 Nat. Commun. (SI Fig. 4, see below). Using confocal microscopy on YFP-tagged RGC1/2 we showed distinct localization of YFP-fluorescence with NeoR/RGC1 forming large intracellular clusters, while NeoR/RGC2 are more distributed over the cell. Despite the different cellular distribution, YFP- and NeoR fluorescence strongly colocalize for both approaches, indicating the formation of heterodimeric complexes. Thus, we are very confident that the fluorescence of NeoR provides a solid indication of the localization of the heterodimeric complexes in native zoospores.

Fig. 4. Confocal Images of RGC1-YFP/NeoR and RGC2-YFP/NeoR complexes in ND7/23 cells. (a) RGC1/NeoR complexes in ND7/23 cells: i) DIC Image of transfected cells. ii) scatter plot of joint relationships between RGC1-YFP and (intrinsic) NeoR fluorescence calculated within the region of interest (ROI: see cycle in iv) from images iii) and iv); PCC: Pearson's correlation coefficient. iii) RGC-YFP fluorescence (green) and iv) NeoR fluorescence (red) before bleaching of the ROI (cycle in iv)). v) + vi) RGC-YFP fluorescence (green) and NeoR fluorescence (red) after bleaching of the ROI with 640 nm light. vii) + viii) RGC-YFP fluorescence (green) and NeoR fluorescence (red) after recovery of NeoR fluorescence by illumination with 405 nm light (scale bar, 100 μ m). (b) RGC2/NeoR complexes in ND7/23 cells: images i)-viii) analog to (a). The reversible bleaching of NeoR fluorescence in single cells was repeated in 2 independent experiments using 5 fields of view each.

Fig. S4 taken from Broser et al., 2020, Nat. Commun. Supplementary Information

We now emphasize this in the revised version of the manuscript on page 13/14:

“Given that RGC1 or 2 are only functional in conjunction with NeoR, we propose that NeoR associates with either RGC1 or 2 by forming heterodimeric complexes in fungal zoospores. Supporting this, NeoR strongly colocalizes with RGC1 or 2 when recombinantly expressed in mammalian cells (Broser et al, 2020).“

A second problem is the functional assay - the authors should convince the reader, that the results obtained with protein expressed in mammalian cells can be transferred to the protein function in the fungus?

The cyclase core enzymes of class III NTP cyclases are highly conserved between virtually all organisms from bacteria up to mammals and the canonical mutations that altered substrate specificity between ATP and GTP in these enzymes have been reported for proteins from various biological sources and expression hosts (including homodimeric RGCs from fungi expressed in mammalian cells, insect cells, xenopus oocytes, *C. elegans* and *E. coli* (as shown in Sunahara et al., 1998 JBC (<https://doi.org/10.1074/jbc.273.26.16332>); Scheib et al., 2018 Nat. Commun. (<https://doi.org/10.1038/s41467-018-04428-w>); Henss et al., 2021 BJP (<https://doi.org/10.1111/bph.15445>)). Indeed, the functional asymmetry of the two pseudo-symmetric active sites is also observed using the isolated protein produced in *E. coli* (see. Fig. S4), indicating that this is an intrinsic feature of the heterodimeric system and independent from the host species. We now emphasize that finding on page 12:

"The functional asymmetry of the pseudosymmetric active sites is also evident in RGC1-cat and NeoR-cat (Fig.S4), presenting it as an inherent property of the heterodimeric cyclase core. Thus, this asymmetry likely applies to the native protein complex in the fungus as well."

Page 7: The authors detect high fluctuation in the density of 3,417 {plus minus} 1,543 per zoospore (N = 23). The authors should discuss this. What is the biological reason for this high deviation?

Strictly speaking, the biological reason for this large variation is unknown. However, since we are experimenting with living zoospores, they might be of different ages, or/and have previously been exposed to different light intensities due to inhomogeneous shading, a variation of this range for the photoreceptor content is not so unlikely. Inherent variability is also reflected in the RNA levels extracted by Laundon (2022) (see Fig. 3D).

To account for this, we added the following sentence to the Figure legend of Fig. 3Ciii:

"The variation likely arises from differences in zoospore maturation under slightly varying microenvironments."

Fig. 3E. The authors should take care about reproducibility. There is missing information: How were the different stages of the chytrid lifecycle obtained. A description of the culture conditions, immobilization, and further important aspects should be incorporated in the Methods part.

Our culturing conditions for *R. globosum* are described in method sections on page 16, and according to the reviewers' suggestions we added the missing information about imaging of different life cycle stages in the confocal microscopy section on page 17. We captured different life cycle stages from an unsynchronized liquid culture. The zoospores were immobilized using two approaches: 1. PBS buffer contained > 100 mM NaCl which suppresses zoospore motility. 2. Floating movement was restricted using 0.25% low melting agarose for confocal microscopy and 1% agarose for SMLM.

On page 17 we include now the paragraph:

"Salt concentrations exceeding 100 mM inhibit zoospore motility. To capture images of each life cycle stage (excluding zoospores), 5 ml of PmTG medium was inoculated with a single *R. globosum* colony and incubated for 4-5 days prior to imaging. This unsynchronized liquid culture contains individuals of various life cycle stages."

"To our knowledge, this is the first naturally occurring photosystem that can be measured in vivo, with fluorescence microscopy providing structural detail well below conventional resolution limits." As this is a crucial finding of the manuscript, supporting data should be clearly elaborated. While Figure S1 shows the measurement of precision, the explanation needs improvement to allow reproduction. To improve in page 14/15 the following questions should be addressed: What was the acquisition time of each frame? What was the cut off to distinguish between NeoR signal and autofluorescence. How exactly was the localization precision defined? The mentioned custom python scripts should be made available via a public repository like eg. Github.

We agree that the significance of this finding warrants additional clarification. In response, we have revised the text accompanying Figure S1 to provide a clearer explanation of our measurement precision. Additionally, we elaborate on the underlying assumptions and potential limitations of our approach, ensuring that readers can better interpret the data. We

improved the Figure caption of Figure S1, also citing the relevant paper and software package that was used to analyze the data and furthermore improved the methods section, now including the frame integration on page 18. The python code for rumposome modelling is available at github (see Data Availability).

Overall statistical tests and values should be given to all figures presented.

Statistical tests (unpaired student's t) are added, and values are given in figure legends or in the source data.

Finally, the discussion requires revision, as not all results are addressed. Especially, results regarding single molecule microscopy are not considered. In addition, a conclusion or at least one or two conclusive sentences would be beneficial.

In the discussion, we stated that SMLM allows us to estimate the numbers of NeoR molecules per zoospore and subsequently we compare this number with that of channelrhodopsins found in the eye-spot of *C. reinhardtii*. The lower number of functional RGCs allows us to speculate about an amplification mechanism due to second messenger generation. However, we now also included that the 30 nm pattern deduced from SMLM suggests that the photoreceptor is embedded into the membrane tubing constituting the rumposome.

Addressing the reviewer's suggestion, we added the following paragraph to the discussion (page 15):

"It is known that proton-pumping microbial rhodopsins arrange themselves in 2D lattices when densely packed in the membrane (Müller et al, 2000; Henderson et al, 1990). Yet, the 30 nm spacing observed in SMLM is too large to correspond to packed and distributed dimeric rhodopsin complexes. Instead, we attribute this pattern to the embedding of the photosensors in the hexagonal arranged membrane tubes that constitute the rumposome."

We added a conclusive sentence at the end of the discussion:

"Our data highlight the remarkable diversity of rhodopsin-based light perception shaped by biological evolution and demonstrate super-resolution microscopy on native fluorescent rhodopsin to uncover details of the light-sensing machinery in chytrid zoospores."

- minor concerns that should be addressed

In the introduction, the authors should also mention in the paper regarding phototaxis experiments with zoospores of *Allomyces reticulatus* (Saranak & Foster, 1997).

This is an important point since Saranak & Foster made the first observation that a rhodopsin mediates zoospore phototaxis. Thus, we changed the paragraph in the introduction on page 3 and included the paper.

Page 3: The terminology of rhodopsin guanylyl cyclases is not consistent in the literature. Is there a reason why the abbreviation changed from RhoGC, via RhGC to RGC? Authors should try to use the abbreviation used before or to explain their choice.

The abbreviation for this group of photoreceptors varies among research groups. With the discovery of a large family of heterodimeric rhodopsin guanylyl cyclases in 2020, we opted to simplify the naming to RGC. When studying multiple species, we specify the genus and species using letter prefixes—for example, RGC1 from *R. globosum* is denoted as RgRGC1,

while *R. hyalinum* would be RhRGC1. However, in this study, we use the simplified notation since it focuses exclusively on proteins from *R. globosum*.

Page 5. "Since chytrid zoospores are considered translationally inactive, the presence of RNA may result from inherited and inactive mRNA-loaded ribosomes (Medina et al, 2020)." This sentence needs rewriting if the authors want to highlight that the RNA is of maternal origin as transcription does not take place in zoospores.

We corrected the mistake; spores are transcriptionally and translationally inactive according to Medina et al (2020). Thus, we changed the sentence on page 6 to:

"Since chytrid zoospores are transcriptionally inactive, the detected RNA likely originates from maternally inherited, inactive mRNA-loaded ribosomes"

"The lack of amplification products indicates that each gene is transcribed individually". As PCR tests with fungal gDNA tend to fail, I suggest the authors should use a more cautious way like "suggests", or "could be due to individual transcription of... "

We changed it accordingly.

„A1-retinal" Better use the expression all-trans-retinal which is quite more frequent in the rhodopsins field. Also "A2-(3-hydro-)retinal" should be adapted accordingly

We changed A1-retinal to all-trans retinal and A2-(3-hydro-)retinal to 3,4-dehydro-retinal.

Page 8. "[.] two putative cGMP-gated ion channels (Figure S1)". Figure S1 should be replaced with Figure S2

done

Figure 2 c. It would be nice enlarge this figure to see the described effects more clearly. If an additional figure is allowed, a single figure should be considered.

We enlarged Fig. 2c as suggested.

In figure 3Ciii the graphics is damaged and should be repaired

In our manuscript file Fig. 3Ciii is not damaged, but we will ensure that the graphics are displayed correctly.

- any additional non-essential suggestions for improving the study (which will be at the author's/editor's discretion)

The paragraph "A functional asymmetry in substrate...". If the reader is not an expert working with these proteins, it is difficult to understand, why each respective mutation was produced. To avoid confusion the whole section needs further improvement regarding explanation of the respective mutations etc.

We rephrased the paragraph to facilitate the understanding for non-expert readers.

Dr. Matthias Jan Broser
Humboldt Universität zu Berlin
Experimental Biophysics
Invalidenstrasse 42
Berlin, Berlin 10115
Germany

11th Apr 2025

Re: EMBOJ-2025-120252R
Super-Resolution Imaging of Native Fluorescent Photoreceptors in Chytrid Fungal Eyes

Dear Dr. Broser,

Thank you for submitting your revised manuscript to The EMBO Journal. Two of the original referees have now assessed it once more, and I am happy to say that both were fully satisfied with the revisions and have no further concerns at this stage. After incorporation of the following remaining editorial issues, we should therefore be able to proceed with formal acceptance of the study:

- Please enter valid email addresses for all coauthors in the submission system, so that they could be informed about the submission and final decision. At resubmission, acknowledgement emails failed to be delivered to Simon Kelterborn.
- Please correct the Section order as follows: Title page - Abstract & Keywords - Introduction - Results - Discussion - Methods - Data Availability - Acknowledgements - Disclosure and Competing Interests Statement - References - Figure Legends - Table(s) - Expanded View Figure Legends.
- Please remove all revision mark-ups from main and Appendix text at this point.
- Please carefully adjust the nomenclature for all main and supplemental figures and tables. Appendix Figures and Tables should be named "Appendix Figure/Table S1..." not just in the Appendix table of contents, but also in their figure legends and all respective in-text call-outs. Expanded View Figures need to be named "Figure EV1..." both in their legends and all in-text call-outs. Also check that all call-outs in the text come up sequentially in numerical/alphabetical order, and that each figure panel or table is referenced at least once. Currently, references to the EV figures and to Appendix Table S3 appear to be missing.
- Please remove the Reagents and Tool table from the main manuscript text - it should only remain as a separately uploaded file.
- As we are switching from a free-text author contribution statement towards a more formal statement based on Contributor Role Taxonomy (CRediT) terms, please remove the present Author Contribution section and instead specify each author's contribution(s) directly in the Author Information page of our submission system during upload of the final manuscript. See <https://casrai.org/credit/> for more information.
- Please correct the format of the reference list according to EMBO Journal guidelines (see <https://www.embopress.org/page/journal/14602075/authorguide#referencesformat> for detail), and make sure that all citations are complete with up to 10 first authors, year, journal abbreviation, volume and page number/eLocator. DOIs or hyperlinks should not be included, except for pre-publications that do not have a pagination yet.
- We still need you to complete and upload the Source Data Checklist that had been sent to you by our Source Data curator, Hannah Sonntag (I am attaching it once more to this message). Please also double-check to make sure that all requested Source Data items have been uploaded. Finally, please note that Source Data files need to be saved according to a scheme: one figure per folder, zipping of the folder, and then uploaded as .zip files. E.g. all the Source data files for figure 1 need to be saved in a single folder and this needs to be zipped and then uploaded as "SD figure 1.zip" file. Only for Expanded View and/or Appendix figures, ZIP together all source data.
- Finally, during routine pre-acceptance checks, our data editors have raised the following queries regarding figures, data, and legends, which I would ask you to address (ideally using the Track Changes option):
 1. Please provide the exact p values in the legends of figures 3B, D; 4B, D; 5B, S3 C, S4.
 2. Please note that in figures S4 there is a mismatch between the annotated p values in the figure legend and the annotated p values in the figure file, which should be corrected.
 3. Please note that the box plots need to be defined in terms of minima, maxima, centre, bounds of box and whiskers, and percentile in the legends of figures 3D, S2.
 4. Please note that information related to N is missing in the legend of figure S3 A.

5. Please note that the measure of center for the error bars needs to be defined in the legends of figures S3 A, B, C; S4

I am returning the manuscript to you for a final round of revision, solely to allow you to make these modifications and upload the revised files. Once we will have received them, we should be ready to swiftly proceed with formal acceptance and production of the manuscript.

With kind regards,

Hartmut

9) To facilitate reproducibility and cross-laboratory adoption of methodologies, please structure the Materials & Methods section as outlined in our guide to authors, including a completed Reagents and Tools Table that can be downloaded from our author guidelines as well (<https://www.embopress.org/page/journal/14602075/authorguide#structuredmethods>).

10) Digital image enhancement is acceptable practice, as long as it accurately represents the original data and conforms to community standards. If a figure has been subjected to significant electronic manipulation, this must be clearly noted in the figure legend and/or the 'Materials and Methods' section. The editors reserve the right to request original versions of figures and the original images that were used to assemble the figure. Finally, we generally encourage uploading of numerical as well as gel/blot image source data; for details see: embopress.org/page/journal/14602075/authorguide#sourcedata

Further information is available in our Guide For Authors:

In the interest of ensuring the conceptual advance provided by the work, we recommend submitting a revision within 3 months (10th Jul 2025). Please discuss the revision progress ahead of this time with the editor if you require more time to complete the revisions. Use the link below to submit your revision:

Link Not Available

Referee #1:

I appreciate the authors addressing all my comments; therefore I recommend the acceptance of this manuscript for publishing in The EMBO Journal.

Referee #3:

All critical points were sufficiently addressed.